



# Dynamics of deep soil carbon – insights from [14]C time-series across a climatic gradient

Tessa Sophia van der Voort[1], Utsav Mannu[1,†], Frank Hagedorn[2], Cameron McIntyre[1,3],

Lorenz Walthert[2], Patrick Schleppi[2], Negar Haghipour[1], Timothy Ian Eglinton[1]

[1]Institute of Geology, ETH Zürich, Sonneggstrasse 5, 8092 Zürich, Switzerland
[2]Forest soils and Biogeochemistry, Swiss Federal Research Institute WSL, Zürcherstrasse 111, 8903
Birmensdorf, Switzerland
[3]Department of Physics, Laboratory of Ion Beam Physics, ETH Zurich, Schaffmattstrasse 20, 9083 Zurich
[†]New address: Department of Earth and Climate Science, IISER Pune, Pune, India

*correspondence to*: Tessa Sophia van der Voort (tessa.vandervoort@erdw.ethz.ch)

**Abstract.** Quantitative constraints on soil organic matter (SOM) dynamics are essential for comprehensive understanding of the terrestrial carbon cycle. Deep soil carbon is of particular interest, as it represents large stocks and its turnover rates remain highly uncertain. In this study, SOM dynamics in both the top and deep soil across a climatic (average temperature ~1-9 °C) gradient are determined using time-series (~20 years) [14]C data from bulk soil and water-extractable organic carbon (WEOC). Analytical measurements reveal enrichment of bomb-derived radiocarbon in the deep soil layers on the bulk level during the last two decades. The WEOC pool is strongly enriched in bomb-derived carbon, indicating that it is a dynamic pool. A numerical model was constructed to determine turnover time of the bulk, slow and dynamic pool as well as the size of the dynamic pool. The presence of bomb-derived carbon in the deep soil, as well as the rapidly turning over WEOC pool and sizeable dynamic fraction at depth across the climatic gradient implies that there likely is a dynamic component of carbon in the deep soil. Precipitation and bedrock type appear to exert a stronger influence on soil C turnover and stocks as compared to temperature.

## 1    Introduction

Within the broad societal challenges accompanying climate and land use change, a better understanding of the drivers of turnover of carbon in the largest terrestrial reservoir of organic carbon, as constituted by soil organic matter (SOM), is essential (Batjes, 1996; Davidson and Janssens, 2006; Doetterl et al., 2015; Prietzel et al., 2016). Terrestrial carbon turnover remains one of the largest uncertainties in climate model predictions (Carvalhais et al., 2014; He et al., 2016). At present, there is no consensus on the net effect that climate and land use change will have on SOM stocks (Crowther et al., 2016; Gosheva et al., 2017; Melillo et al., 2002; Schimel et al., 2001; Trumbore and Czimczik, 2008). Deep soil carbon is of particular interest because of its large stocks (Jobbagy and Jackson, 2000; Rumpel and Kogel-Knabner, 2011) and perceived stability.  The stability is indicated by low [14]C content (Rethemeyer et al., 2005; Schrumpf et al., 2013; van der Voort et al., 2016) and low microbial activity (Fierer et al., 2003). Despite its importance, deep soil carbon has been sparsely studied and remains poorly understood (Angst et al., 2016; Rumpel and Kogel-Knabner, 2011). The inherent complexity of SOM and the multitude of drivers controlling its stability further impedes the understanding of this globally significant carbon pool (Schmidt et al., 2011). In this framework, there is a particular interest in the portion of soil carbon that could be most vulnerable to change, especially in colder climates (Crowther et al., 2016).





Water-exactable organic carbon (WEOC) is seen as a dynamic and potentially vulnerable carbon pool in the soil
(Hagedorn et al., 2004; Lechleitner et al., 2016). Radiocarbon ($^{14}$C) can be a powerful tool to determine the
dynamics of carbon turnover over decadal to millennial timescales because of the incorporation of bomb-
derived $^{14}$C introduced in the atmosphere in the 1950's as well as the radioactive decay of $^{14}$C naturally present
in the atmosphere (Torn et al., 2009). Time-series $^{14}$C data is particularly insightful because it enables the
tracking of recent decadal carbon. Furthermore, single time-point $^{14}$C data can yield two estimates for turnover
time, whilst time-series data yields a single turnover estimate (Torn et al., 2009). Given that the so-called "bomb
radiocarbon spike" will continue to diminish in the coming decades, time-series measurements are increasingly
a matter of urgency in order to take full advantage of this intrinsic tracer (Graven, 2015). Several case-studies
have collected time-series $^{14}$C soil datasets and demonstrated the value of this approach (Baisden and Parfitt,
2007; Prior et al., 2007; Fröberg et al., 2010; Mills et al., 2013, Schrumpf and Kaiser, 2015). However, these
studies are sparse, based on specific single sites and have been rarely linked to abiotic and biotic parameters.
Much more is yet to be learned about the carbon cycling through time-series observations in top- and subsoils
along environmental gradients. Furthermore, to our knowledge, there are no studies with pool-specific $^{14}$C soil
time-series focusing on labile carbon.

This study assesses two-pool soil carbon dynamics as determined by time-series (~20 years) radiocarbon across
a climatic gradient. The time-series data is analyzed by a numerically optimized model with a robust error
reduction to yield carbon turnover estimates for the bulk, dynamic and slow pool. Model output is linked to
potential drivers such as climate, forest productivity and physico-chemical soil properties. The overall objective
of this study is to improve our understanding of shallow and deep soil carbon dynamics in a wide range of
ecosystems.

**2 Materials and methods**
**2.1 Study sites, sampling strategy and WEOC extraction**
The five sites investigated in this study are located in Switzerland between 46-47° N and 6-10° E and
encompass large climatic (mean annual temperature (MAT) 1.3-9.2°C, mean annual precipitation (MAP) 864-
2126 mm m$^{-2}$y$^{-1}$) and geological gradients (Table 1). The sites are part of the Long-term Forest Ecosystem
Research program (LWF) at the Swiss Federal Institute for Forest, Snow and Landscape Research, WSL
(Schaub et al., 2011; Etzold et al., 2014). The soils of these sites were sampled between 1995 and 1998
(Walthert et al., 2002, 2003) and were re-sampled following the same sampling strategy in 2014 with the aim to
minimize noise caused by small-scale soil heterogeneity. In both instances sixteen samples were taken on a
regular grid on the identical 43 by 43 meters (~1600 m$^2$) plot (Fig. 1; see Van der Voort et al., 2016 for further
details). For the archived samples taken between 1995 and 1998, mineral soil samples down to 40 cm depth
(intervals of 0-5, 5-10, 10-20 and 20-40 cm) were taken on an area of 0.5 by 0.5 m (0.25 m$^2$). For samples >40
cm (intervals of 40-60, 60-80 and 80-100 cm), corers were used to acquire samples (n=5 in every pit, area
~2.8×10$^{-3}$ m$^2$). The organic layer was sampled by use of a metal frame (30×30 cm). The samples were dried at
35-40°C, sieved to remove coarse material (2 mm), and stored in hard plastic containers under controlled
climate conditions in the "Pedothek" at WSL (Walthert et al., 2002). For the samples acquired in 2014 the same
sampling strategy was followed, and samples were taken on the exact same plot proximal (~10 m) to the legacy





samples. For the sampling, a SHK Martin Burch AG HUMAX soil corer ($\sim 2 \times 10^{-3}$ m$^2$) was used for all depths
(0-100 cm). For the organic layer, a metal frame of 20×20 cm was used to sample. Samples were sieved (2 mm),
frozen and freeze-dried using an oil-free vacuum-pump powered freeze dryer (Christ, Alpha 1-4 LO *plus*). For
the time-series radiocarbon measurements, all samples covering $\sim 1600$ m$^2$ were pooled to one composite sample
per soil depth using the bulk-density. In order to determine bulk-density of the fine earth of the 2014 samples,
stones > 2 mm were assumed to have a density of 2.65 g/cm$^3$. For the Alptal site, sixteen cores were taken on a
slightly smaller area ($\sim 1500$ m$^2$) which encompasses the control plot of a nitrogen addition experiment
(NITREX project) (Schleppi et al., 1998). For this site, no archived samples are available and thus only the 2014
samples were analyzed. Soil carbon stocks were estimated by multiplying SOC concentrations with the mass of
soil calculated from measured bulk densities and stone contents for each depth interval (Gosheva et al., 2017).
For the Nationalpark site, the soil carbon stocks from 80-100 cm were estimated using data from a separately
dug soil profile (Walthert et al., 2003) because the HUMAX corer could not penetrate the rock-dense soil below
80 cm depth. In order to understand very deep soil carbon dynamics (i.e. >100 cm), this study also includes
single-time point $^{14}$C analyses of soil profiles that were dug down to the bedrock between 1995 and 1998 as part
of the LWF programme on the same sites (Walthert et al., 2002). The sampling of the profiles has not yet been
repeated.

**2.2 Climate and soil data**
Temperature and precipitation data are derived from weather stations close to the study sites that have been
measuring for over two decades, yielding representative estimates of both variables and over the time period
concerned in this study (Etzold et al., 2014). The pH values for all sites and concerned depth intervals were
acquired during the initial sampling campaign (Walthert et al. 2002). At Alptal, pH values were determined as
described in Xu et al. (2009), values of 10-15 cm were extrapolated to the deeper horizons because of the
uniform nature of the Gley horizon. Exchangeable cations were extracted (in triplicate) from the 2-mm-sieved
soil in an unbuffered solution of 1 M NH$_4$Cl for 1 hour on an end-over-end shaker using a soil-to-extract ratio of
1:10. The element concentrations in the extracts were determined by inductively coupled plasma atomic
emission spectroscopy (ICP-AES) (Optima 3000, Perkin–Elmer). Contents of exchangeable protons were
calculated as the difference between the total and the Al-induced exchangeable acidity as determined (in
duplicate) by the KCl method (Thomas, 1982). This method was applied only to soil samples with a pH (CaCl$_2$)
< 6.5. In samples with a higher pH, we assumed the quantities of exchangeable protons were negligible. The
effective cation-exchange capacity (CEC) was calculated by summing up the charge equivalents of
exchangeable Na, K, Mg, Ca, Mn, Al, Fe and H. The base saturation (BS) was defined as the percental fraction
of exchangeable Na, K, Mg, and Ca of the CEC (Walthert et al., 2002, 2013). Net primary production (NPP)
was determined by Etzold et al. (2014) as the sum of carbon fluxes by woody tree growth, foliage, fruit
production and fine root production. Soil texture (sand, silt and clay content) on plot-averaged samples taken in
2014 have been determined using grain size classes for sand, silt and clay respectively of 0.05-2 mm, 0.002-0.05
mm and <0.002 mm according to Klute (1986). The continuous distribution of grain sizes was also determined
after removal of organic matter (350 °C for 12 h) using the Mastersizer 2000 (Malvern Instruments Ltd.). Soil
water potential (SWP) was measured on the same sites as described in Von Arx et al., (2013).





### 2.3 Isotopic ($^{14}$C, $^{13}$C) and compositional (C, N) analysis

Prior to the isotopic analyses, inorganic carbon in all samples was removed by vapour acidification for 72 hours (12M HCl) in desiccators at 60 °C (Komada et al., 2008). After fumigation, the acid was neutralised by substituting NaOH pellets for another 48 hours. All glassware used during sample preparation was cleaned and combusted at 450°C for six hours prior to use. Water extractable organic carbon (WEOC) was procured by extracting dried soil with of 0.5 wt% pre-combusted NaCl in ultrapure Milli-Q (MQ) water in a 1:4 soil:water mass ratio (adapted from Hagedorn et al., (2004), details in Lechleitner et al., (2016)).

In order to determine absolute organic carbon and nitrogen content as well as $^{13}$C values, an Elemental Analyser-Isotope Ratio Mass Spectrometer system was used (EA-IRMS, Elementar, vario MICRO cube – Isoprime, Vison). Atropine (Säntis) and an in-house standard peptone (Sigma) were used for the calibration of the EA-IRMS for respectively carbon concentration, nitrogen concentration and C:N ratios and $^{13}$C. High $^{13}$C values were used to flag if all inorganic carbon had been removed by acidification.

For the $^{14}$C measurements of the bulk soil samples were first graphitised using an EA-AGE (elemental analyser-automated graphitization equipment, Ionplus AG) system at the Laboratory of Ion Beam Physics at ETH Zürich (Wacker et al., 2009). Graphite samples were measured on a MICADAS (MIniturised radioCArbon DAting System, Ionplus AG) also at the Laboratory of Ion Beam Physics, ETH Zürich (Wacker et al., 2010). For three samples (Alptal depth intervals 40-60, 60-80 and 80-100 cm) the $^{14}$C signature was directly measured as $CO_2$ gas using the recently developed online elemental analyzer (EA) - stable isotope ratio mass spectrometers (IRMS)–AMS system et ETH Zürich (McIntyre et al., 2016). Oxalic acid (NIST SRM 4990C) was used as the normalising standard. Phthalic anhydride and in-house anthracite coal were used as blank. Two in-house soil standards (Alptal soil 0-5 cm, Othmarsingen soil 0-5 cm) were used as secondary standards. For the WEOC, samples were converted to $CO_2$ by Wet Chemical Oxidation (WCO) (Lang et al., 2016) and run on the AMS using a Gas Ion Source (GIS) interface (Ionplus). To correct for contamination, a range of modern standards (sucrose, Sigma, δ13C = -12.4 ‰ VPDB, F$^{14}$C = 1.053 ± 0.003) and fossil standards (phthalic acid, Sigma, δ13C = -33.6‰ VPDB, F$^{14}$C <0.0025) were used (Lechleitner et al., 2016).

### 2.4 Numerical optimization turnover and vulnerable fraction

### 2.4.1 Time-series based determination of likeliest turnover time

In order to optimally constrain carbon turnover estimates for the $^{14}$C time-series data, a numerical model was constructed in MATLAB version 2015a (The MathWorks, Inc., Natick, Massachusetts, United States). For the turnover estimation, we assumed the system to be in steady state over the modeled period (~$1\times10^4$ years, indicating soil formation since the last glacial retreat (Ivy-Ochs et al., 2009)), hence accounting both for radioactive decay and incorporation of the bomb-testing derived material produced in the 1950's and 1960's (Eq. 1.) (Herold et al., 2014; Torn et al., 2009).

$$R_{sample,t} = k \times R_{atm,t} + (1 - k - \lambda) \times R_{sample(t-1)} \qquad (1)$$

$$R_{sample,t} = \frac{\Delta^{14}C_{sample}}{1000} + 1 \qquad (2)$$





In Eq. 1-2, the constant for radioactive decay of $^{14}C$ is indicated as $\lambda$, the decomposition rate $k$ (inverse of
turnover time) is the only unknown in this equation and is hence the variable for which the optimal value that
fits the data is sought using the model. The R value of the sample is inferred from $\Delta^{14}C$, hence accounting for
the sampling year, as shown in Eq. (2) (Herold et al., 2014; Solly et al., 2013). In order to avoid ambiguity the
term *turnover time* and not i.e. mean residence time is used solely in this manuscript (Sierra et al., 2016). For
computation of the optimal turnover time we assumed an initial fraction modern ($F_m$) of $^{14}C$ value of 1 at 10000
B.C.. For the period after 1900 atmospheric fraction modern ($F_m$) values of the Northern Hemisphere were used
(Levin et al., 2010).
Using Eq. 1-2, we then computed the $R_{sample,t}$ for two given time points (1995-1998 A.D., depending on the
year of initial sampling and 2014 AD) within a range of turnover time of 1-10000 years. The exhaustive
numerical optimization evaluates the likelihood of every single solution (precision to 0.1 year) and yields the
turnover rate which is the optimal fit for the two data points (Fig. 2 and 3). In order to account for vegetation-
lag, two scenarios were run: firstly (1) with no assumed lag between the fixation of carbon from the atmosphere
and input into to the soil and (2) model run with a lag of fixation of the atmospheric carbon as inferred from the
dominant vegetation (Von Arx et al., 2013; Etzold et al., 2014). In the case of full deciduous trees coverage a
lag of two years was assumed, and for the case of 100% conifer-dominated coverage a lag of 8 years was
incorporated (Table 1). Turnover times determined with the numerical optimization match the manually
optimized turnover modeling published previously (Herold et al., 2014; Solly et al., 2013).
**2.4.2   Turnover and size vulnerable pool based on two-pool model**
As SOM is complex and composed of a continuum of pools with various ages (Schrumpf and Kaiser, 2015) and
there is data available from two SOM pools, a two-pool model was created (Fig. 3). WEOC constitutes only a
small portion of the total carbon (<1%), but could be representative for a larger component of rapidly turning
over carbon, even in the deep soil (Baisden and Parfitt, 2007; Koarashi et al., 2012). Using the data from the
bulk soil and WEOC time-series, the turnover of the slow pool and the relative size of the dynamic pool can be
determined. The following assumption were made: First, both pools (slow & fast) make up the total carbon pool
(Eq. 3). Secondly, the total turnover of the bulk soil is made up out of the "dynamic" fraction turnover
multiplied by "dynamic" fraction pool size and the "slow" pool turnover multiplied by "slow" pool size (Eq. 4).
Lastly, we assume that the signature of the sample (the time-series bulk data) is determined by the rate of
incorporation of the material (atmospheric signal) and the loss of carbon the two pools (Eq. 5).
$$1 = F_1 + F_2 \quad (3)$$
$$k_{total} = (k_1 \cdot F_1) + (k_1 \cdot F_2) \quad (4)$$
$$R_{sample,t} = k_{total} \times R_{atm,t} + F_1[(1 - k_1 - \lambda) \times R_{sample(t-1)}] + F_2(1 - k_2 - \lambda) \times R_{sample(t-1)} \quad (5)$$
Where $F_1$ is the relative size of the dynamic pool, and $F_2$ is the relative size of the (more) stable pool. The $k_1$ is
the inverse of the turnover time of the WEOC as determined using the numerical optimisation of Eq. (1) and (2),
and $k_1$ is determined by numerical optimisation. The $k_2$ is the inverse of the turnover of the slow pool. The




numerical optimization finds the likeliest solution for the given dataset. Further details can be found in the
Supplementary Information (SI) text and SI Fig. 1. Due to limited availability of archived samples, there are
only single time points available for some samples as indicated in Fig. 4. The Matlab-based numerical
optimization code will be made available upon publication. For correlations (packages HMISC, corrgram,
method = pearson, significance $p<0.05$), statistical software R version 1.0.153 was used.

**3 Results**
**3.1 Changes of radiocarbon signatures over time**
Overall, there is a pronounced decrease in radiocarbon signature with soil depth at all sites (Fig. 4). The time-
series results show clear changes in radiocarbon signature over time from the initial sampling period (1995-
1998) as compared to 2014, with the magnitude of change depending on site and soil depth. In the uppermost 5
cm of soils, the overarching trend in the bulk soil is a decrease in the $^{14}$C bomb-spike signature in the warmer
climates (Othmarsingen, Lausanne), whilst at higher elevation (colder) sites (Beatenberg, Nationalpark) the
bomb-derived carbon appears to enter the top soil between 1995-8 and 2014.

207          Water-extractable OC (WEOC) has an atmospheric $^{14}$C signature in the top soil at all sites in 2014. The
deep soil in the 1990's still has a negative $\Delta^{14}$C signature of WEOC at multiple sites. There are two
distinguishable types of depth trends for WEOC in the 2014 dataset: (1) WEOC has the same approximate $^{14}$C
signature throughout depth (Othmarsingen, Beatenberg), (2) WEOC becomes increasingly $^{14}$C depleted with
depth (Alptal, Nationalpark), or an intermediate form where WEO$^{14}$C is modern throughout the top soil but
becomes more depleted of $^{14}$C in the deep soil (Lausanne) (Fig. 4). The isotopic trends of WEOC co-vary with
grain size as inherited from the bedrock type (Walthert et al., 2003). Soils with a relatively modern WEO$^{14}$C
signature in 2014 (down to 40 cm) are underlain by bedrock with large grained (SI Fig. 3, Table SI 3)
components (the moraines and sandstone at Othmarsingen, Lausanne and Beatenberg respectively). Soils where
WEO$^{14}$C signature decreases with depth are underlain by bedrock containing fine-grained components. For
instance, the Flysch in Alptal (Schleppi et al., 1998) and intercalating layers of silt and coarse grained alluvial
fan in Nationalpark (Walthert et al., 2003) respectively.

**3.2 Carbon turnover and the dynamic fraction**
Incorporation of a vegetation-induced time lag (SI Fig. 2, Table 2) has an effect on modelled carbon dynamics
in the organic layer, but this effect is strongly attenuated in the 0-5 cm layer in the mineral soil and virtually
absent for the deeper soil layers. Turnover times show two modes of behavior for well-drained soils and
hydromorphic soils, respectively. The non-hydromorphic soils have relatively similar values with decadal
turnover times for the 0-5 cm layer, increasing to an order of centuries down to 20 cm depth, and to millenia in
deeper soil layers (980 to 3940 years at 0.6 to 1 m depth) (Fig. 5). In contrast, the hydromorphic soils are
marked by turnover times that are up to an order of magnitude larger, from centennial in top soil to (multi-)
millennial in deeper soils. At the Beatenberg podsol, turnover time of the deepest layer (40-60 cm, ~1900 y) is
faster than the shallow layer (20-40 cm, ~1300 y) (Figure 5, SI Table 5).
Carbon stocks also show distinct difference between drained and hydromorphic soils with greater stock in the
hydromorphic soils (~15 kg C m$^{-2}$ at Beatenberg and Alptal vs. ~ 6 - ~7 kg C m$^{-2}$ at Othmarsingen, Lausanne
and Nationalpark, Fig. 5, Table 3)).



The turnover times of the WEOC mimic the trends in the bulk soil but are up to an order of magnitude
faster. Considering WEOC turnover in the non-hydromorphic soils only, there is a slight increase in WEOC
turnover with decreasing site temperature, but the trend is not significant. The modeled dynamic faction is
sizeable at the surface but decreases towards the lower top soil (from ~0.4 at 0-5 cm to ~0.1 at 10-20 cm in
Othmarsingen). In the deep soil, there is also a non-negligible proportion of dynamic carbon (e.g. 0.14-0.46 at
20-40 cm).

**3.3 Pre-glacial carbon in deep soil profiles**
The turnover times of deep soil carbon exceed 10,000 years in several profiles, indicating the presence of carbon
that pre-dates the glacial retreat (Fig. 6). These profiles are located on carbon-containing bedrock and concern
the deeper soil (80-100 cm) of the Gleysol (Alptal), as well as >100 cm in the Cambisol (Lausanne) (Fig. 6).

**3.4 Environmental drivers of carbon dynamics**
Pearson correlation was used to assess potential relationships between carbon stocks, turnover and fluxes and
their potential controlling factors (climate, NPP, soil texture, soil moisture and physicochemical properties
(Table 4)). For the averaged top soil (0-20 cm, n=5), carbon stocks were significantly positive correlated to
Mean Annual Precipitation (MAP). Turnover time in the bulk top soil negatively correlated with silt content and
positively with average grain size. Turnover time in the WEOC of the top soil did not correlate significantly
with any parameter. The modeled dynamic soil fraction in the top soil does positively correlate with MAP and
clay content and negatively with sand content. Deeper soil bulk stock and turnover positively correlated with
MAP and negatively with Cation Exchange Capacity (CEC). For non-hydromorphic sites, the fraction of
dynamic carbon increases with decreasing MAT at all depths, but the trend is not significant (e.g. from ~0.4 -
~0.9 at 0-5 cm).

**4 Discussion**
**4.1 Dynamic deep soil carbon**
**4.1.1   Rapid shifts in $^{14}$C abundance reflect dynamic deep carbon**
The propagation of bomb-derived carbon into supposedly stable deep soil on the bulk level across the climatic
gradient implies that SOM in deep soil contains a dynamic pool and could be less stable and potentially more
vulnerable to change than previously thought. This possibility is further supported by the WEO$^{14}$C which is
consistently more enriched in bomb-derived carbon than the bulk soil. Near-atmospheric signature WEO$^{14}$C
pervades up to 40 or even 60 cm depth. Hagedorn et al., (2004) also found WEOC to be a highly dynamic pool
using $^{13}$C tracer experiments in forest soils.
We consider our $^{14}$C comparison over time to be robust because the grid-based sampling and averaging was
repeated on the same plots which excludes the effect plot-scale variability (Van der Voort et al., 2016). Our $^{14}$C
time-series data in the deep soil corroborate pronounced changes in $^{14}$C (hence substantial SOM turnover) in
subsoils of an area with pine afforestation (Richter and Markewitz, 2001). The findings are also in agreement
with results from an incubation study by Fontaine et al., (2007) which showed that the deep soil can have a



significant dynamic component. Baisden et al., (2007) also found indications of a deep dynamic pool using
modeling on [14]C time-series on the bulk level on a New Zealand soil under stable pastoral management.

**4.1.2    Carbon dynamics reflect soil-specific characteristics at depth**
Bulk carbon turnover for the top and deeper soil fall in the range of prior observations and models, although the
data for the latter category is sparse (Scharpenseel and Becker-Heidelmann, 1989; Paul et al., 1997; Schmidt et
al., 2011; Mills et al., 2013; Braakhekke et al., 2014).   The carbon turnover is related to soil-specific
characteristics. The slower turnover of hydromorphic as compared to non-hydromorphic soils is likely due to
increased waterlogging and limited aerobicity (Hagedorn et al., 2001) which is conducive to slow turnover and
enhanced carbon accumulation. The WEOC turns over up to an order of magnitude faster than the bulk and
mirrors these trends, indicating that it indeed is a more dynamic pool (Hagedorn et al., 2004; Lechleitner et al.,
2016). Results also reflect known horizon-specific dynamics for certain soil types, particularly in the deep soil.
The hydromorphic Podsol at Beatenberg shows specific pedogenetic features such as an illuviation layer with an
enrichment in humus and iron in the deeper soil (Walthert et al., 2003) where turnover of bulk and WEOC is
faster and stocks are higher than in the elluvial layer above (Fig. 5). This is likely due to the input of younger
carbon via leaching of dissolved organic carbon. The non-hydromorphic Luvisols are marked by an enrichment
of clay in the deeper soil, which can enhance carbon stabilization (Lutzow et al., 2006). This also reflected in
the turnover time of the 60-80 cm layer in the Othmarsingen Luvisol – in this clay-enriched depth interval
(Walthert et al., 2003), turnover is relatively slow as compared to the other (colder) non-hydromorphic soils
(Fig. 5).

**4.1.3    Sizeable dynamic pool at depth & implications for carbon transport**
The [14]C time series modelling indicate that the size of the dynamic pool can be large, even at greater depth than
it was observed by other [14]C time-series (Richter and Markewitz, 2001; Baisden and Parfitt, 2007; Koarashi et
al., 2012). The two-pool modelling indicates that the size of dynamic pool in the deep soil can be upwards of
~14%. A deep dynamic pool is consistent with findings of a [13]C tracer experiment by Hagedorn et al., (2001)
that shows with that relatively young (<4 years) carbon can be rapidly incorporated in the top soil (20% new C
at 0-20 cm depth) but also in the deep soil (50 cm). In our study, the illuvial horizon of the Podzol stands out
again with a higher amount of the dynamic fraction than the elluvial horizon above. Rumpel and Kögel-Knabner
(2011) have highlighted the importance of the poorly understood deep soil carbon stocks and a significant
dynamic pool in the deep soil could imply that carbon is more vulnerable than initially suspected. One major
input pathway of younger C into deeper soils is the leaching of DOC (Kaiser and Kalbitz, 2012; Sanderman and
Amundson, 2009). Here, we have measured WEOC – likely primarily composed of microbial metabolites
(Hagedorn et al., 2004) – carrying a younger [14]C signature than bulk SOM and thus, representing a translocator
of fresh carbon to the deep soil. In addition to WEOC, roots and associated mycorrhizal communities may also
provide a substantial input of new C into soils in deeper soils (Rasse et al., 2005). Considering the non-
hydromorphic soils alone, the size of the dynamic pool increases with site elevation and cooler MAT. This is
consistent with findings of Budge et al., (2011) and Leifeld et al., (2009) that grassland soils at higher elevation
have larger labile SOM pools.





### 4.2 Contribution of petrogenic carbon

Our results on deep soil carbon suggest the presence of pre-aged or $^{14}$C-dead (fossil), pre-interglacial carbon in the Alptal (Gleysol) and Lausanne (Cambisol) profiles, implying that a component of soil carbon is not necessarily linked to recent (< millenial) terrestrial productivity and instead constitutes part of the long-term (geological) carbon cycle (> millions of years). In the case of the Gleysol in Alptal, the $^{14}$C-depleted material could be derived from the poorly consolidated sedimentary rocks (Flysch) in the region (Hagedorn et al., 2001a; Schleppi et al., 1998; Smith et al., 2013), whereas carbon present in glacial deposits and molasse may contribute in deeper soils at the Lausanne (Cambisol) site. The potential contribution of fossil carbon was estimated using a mixing model using the signature of a soil without fossil carbon, the signature of fossil carbon and the measured values (SI Table 4). Fossil carbon contribution in the Alptal profile between 80-100 cm (Fig. 6, SI Table 4) is estimated at ~40 %. Below one meter at Lausanne site the petrogenic percentage ranges from ~20% at 145 cm up to ~80 % at 310 cm depth (Fig. 6, SI Table 4).

Other studies analyzing soils have observed the significant presence of petrogenic (geogenic in soil science terminology) in loess-based soils (Helfrich et al., 2007; Paul et al., 2001). Our results suggest that pre-glacial carbon may comprise a dominant component of deep soil organic matter in several cases, resulting in an apparent increase in the average age (and decrease in turnover) of carbon in these soils. Hemingway et al., (2018) have highlighted that fossil carbon oxidized in soils can lead to significant additional $CO_2$ emissions. Therefore, the potential of soils to 'activate' fossil petrogenic carbon should be considered when evaluating the soil carbon sequestration potential.

### 4.3 Controls on carbon dynamics and cycling

In order to examine the effects of potential drivers on soil C turnover, stocks and the size of the dynamic pool, we explore correlations between a number of available factors which have previously been proposed, such as texture, geology, precipitation, temperature and soil moisture (Doetterl et al., 2015; McFarlane et al., 2013; Nussbaum et al., 2014; Seneviratne et al., 2010; van der Voort et al., 2016). The vegetation-induced lag does not strongly impact turnover times except in the organic layer and in the top 5 cm of the mineral soil (SI Fig. 2). From examination of data for all samples it emerges that C turnover does not exhibit a consistent correlation with any specific climatological or physico-chemical factor. This implies that no single mechanism predominates and/or that there is a combined impact of geology and precipitation as these soil-forming factors affect grain size distribution, water regime and mass transport in soils. Exploring potential relationships in greater detail, we see that carbon stocks in the top soil and deep soil as well as turnover time is positively related to MAP, which could be linked to waterlogging and anaerobic conditions even in upland soils leading to a lower decomposition and thus to a higher build-up of organic material (Keiluweit et al., 2015). Our results are supported by the findings based on >1000 forest sites that precipitation exerts a strong effect on soil C stocks across Switzerland (Gosheva et al., 2017; Nussbaum et al., 2014). Turnover in both top and deep soil was most closely correlated with texture. The positive correlation of top soil turnover with grain size and negative correlation with the amount of silt-sized particles reflects lower stabilization in larger-grained soils as opposed to clay-rich soils with a higher and more reactive surface area (Rumpel and Kogel-Knabner, 2011). The modeled size of the dynamic pool is mostly related to precipitation and texture. It correlates positively with MAP and clay content and negatively with sand content. This correlation could be because sandier soils offer



less reactive surfaces for SOM stabilization as opposed to clay-rich soils (Lutzow et al., 2006). Additionally, wetter conditions inhibit SOM breakdown. Overall, geology seems to impact the carbon cycling in three key ways. Firstly, when petrogenic carbon is present in the bedrock from shale or reworked shale (Schleppi et al., 1998; Walthert et al., 2003), fossil carbon contributes to soil carbon. Secondly, porosity of underlying bedrock either prevents or induces waterlogging which in turn affects turnover. Thirdly, the initial components of the bedrock (i.e. silt-sizes layers in an alluvial fan) influence the final grain size distribution and mineralogy (SI Fig. 3, Table 3), which is also reflected in the bulk and pool-specific turnover. Within the limited geographic and temporal scope of this paper, we hypothesize that for soil carbon stocks and their turnover, temperature is not the dominant driver, which has been concluded by some (Giardina and Ryan, 2000) but refuted by others (Davidson et al., 2000; Feng et al., 2008). The only climate-related driver which appears to be significant is precipitation.

### 4.4 Modular robust numerical optimization

The numerical approach used here builds on previous work concerning turnover modeling of bomb-radiocarbon dominated samples (Herold et al., 2014; Solly et al., 2013; Torn et al., 2009) and the approach used in numerous time-series analysis with box modeling using Excel (Schrumpf and Kaiser, 2015) or Excel solver (Baisden et al., 2013; Prior et al., 2007). However certain modifications were made in order to (i) provide objective repeatable estimates, (ii) incorporate the WEOC as a sub-pool of C, and (iii) identify samples impacted by petrogenic (also called geogenic) carbon. Identifying petrogenic carbon in the deep soil is important considering the large carbon stocks in deep soils (Rumpel and Kogel-Knabner, 2011) and the wider relevance of petrogenically-derived carbon in the global carbon cycle (Galy et al., 2008). This approach is modular and could be adapted in the future to identify the correct turnover for time-series $^{14}$C data, which is becoming increasingly important with the falling bomb-peak (Graven, 2015).

### 5 Conclusion

Time-series radiocarbon ($^{14}$C) analyses of soil carbon across a climatic range reveals recent bomb-derived radiocarbon in both upper and deeper bulk soil, implying the presence of a rapidly turning over pool at depth. Pool-specific time-series measurements of the WEOC indicate this is a more dynamic pool which is consistently more enriched in radiocarbon than the bulk. The modeled size of the dynamic fraction is non-negligible even in the deep soil (~0.14-0.46). This could imply that a component of the deep soil carbon could be more dynamic than previously thought.

The interaction between precipitation and geology appears to be the main control on carbon dynamics rather than site temperature. Carbon turnover in non-hydromorphic soils is relatively similar (decades to centuries) despite dissimilar climatological conditions. Hydromorphic soils have turnover times which are up to an order of magnitude slower. These trends are mirrored in the dynamic WEOC pool, suggesting that in sandy, non-waterlogged (aerobic) soils the transport of relatively modern (bomb-derived) carbon into the deep soil and/or the microbial processing is enhanced as compared to fine-grained waterlogged (anaerobic) soils.

Model results indicate certain soils contain significant quantities of pre-glacial or petrogenic (bedrock-derived) carbon in the deeper part of their profiles. This implies that soils not only sequester "modern" but can rather also mobilize and potentially metabolize "fossil" or geogenic carbon.





391    Overall, these time-series $^{14}$C bulk and pool-specific data, coupled to a robust numerical modeling

392  approach, provide novel constraints on soil carbon dynamics in surface and deeper soils for a range of

393  ecosystems.

394



**Acknowledgements**

We would like to acknowledge the SNF NRP68 *Soil as a Resource* program for funding this project (SNF 406840_143023/11.1.13-31.12.15). We would like to thank various members of the Laboratory of Ion Beam Physics and Biogeoscience group for their help with the analyses, in particular Lukas Wacker. We thank Roger Köchli for his crucial help in the field which enabled an effective time-series comparison and for his help with subsequent analyses. We thank Emily Solly and Sia Gosheva for their valuable insights, Claudia Zell for her help on the project and in the field, Peter Waldner for facilitating the fieldwork, and Elisabeth Graf-Pannatier for her insights on soil moisture. The 2014 field campaign would not have been possible without the help of Thomas Blattmann, Lukas Oesch, Markus Vaas and Niko Westphal. Thanks to Stephane Beaussier for the insights into numerical modeling. Also thanks to Nadine Keller and Florian Neugebauer for their help in the lab. Last but not least, thanks to Thomas Bär for summarizing ancillary pH data. Data supporting this paper is provided in a separate data Table.

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



**Author contributions**


T.S. van der Voort planned, coordinated and executed the sampling strategy and sample collection, performed
the analyses, conceptualized and optimized the model and processed resulting data. U. Mannu led the model
development. F. Hagedorn lent his expertise on soil carbon cycling and soil properties. C. McIntyre facilitated
and coordinated the radiocarbon measurements and associated data corrections. L. Walthert and P. Schleppi lent
their expertise on the legacy sampling and provided data for the compositional analysis. N. Haghipour
performed in isotopic and compositional measurements. T. Eglinton provided the conceptual framework and
aided in the paper structure set-up. T.S. van der Voort prepared the manuscript with help of all co-authors.



**Tables**

**Table 1** Overview sampling locations and climatic and ecological parameters.

| Location | Soil type | Geology | Latitude(N)/ Longitude (E) | Soil depth (m) | Depth Upper waterlogging (m)[1] | limit Altitude Elevation (m a.s.l.) | MAT °C | MAP mm y[-1] | NPP g C m[-2]y[-1] |
|---|---|---|---|---|---|---|---|---|---|
| Othmarsingen[1,2,3] | Luvisol | Calcareous moraine | 47°24'/8°14' | >1.9 | 2.5 | 467-500 | 9.2 | 1024 | 845 |
| Lausanne[1,2,3] | Cambisol | Calcarous and shaly moraine | 46°34'/6°39' | >3.2 | 2.5 | 800-814 | 7.6 | 1134 | 824 |
| Alptal[1,2,3,4] | Gleysol | Flysch (carbon-holding sedimentary rock) | 47°02'/8°43' | >1.0 | 0.1 | 1200 | 5.3 | 2126 | 347 |
| Beatenberg[1,2,3] | Podzol | Sandstone | 46°42'/7°46' | 0.65 | 0.5 | 1178-1191 | 4.7 | 1163 | 302 |
| Nationalpark[1,2,3] | Fluvisol | Calcareous alluvial fan | 46°40'/10°14' | >1.1 | 2.5 | 1890-1907 | 1.3 | 864 | 111 |

[1]Walthert et al. (2003) [2]Etzold et al. (2014) [3]Von Arx et al., (2013) [4]Krause et al., (2013) for Alptal data



**Table 2** Vegetation and soil data of the study sites. Soil water potential (hPa) are for 15 cm depth.

| Location | Deciduous tree species (%)[3] | Dominant species[3] | Inferred tree carbon lag fixation (y) | Organic layer Type[1] | Sand (%) | Silt (%) | Clay (%) | Soil water potential (hPa) percentiles[3] | | |
|---|---|---|---|---|---|---|---|---|---|---|
| | | | | | | | | 5% | 50% | 95% |
| Othmarsingen | 100 | *Fagus sylvatica* | 2 | Mull | 47 | 35 | 18 | -577 | -39 | -9 |
| Lausanne | 80 | *Fagus sylvatica* | 3 | Mull | 47 | 34 | 19 | -547 | -49 | -8 |
| Alptal[4] | 15 | *Picea abies* | 7 | Mor to anmoor | 6 | 48 | 46 | -38 | -13 | +1 |
| Beatenberg | 0 | *Picea abies* | 8 | Mor | 86 | 11 | 3 | -50 | -14 | +1 |
| Nationalpark | 0 | *Pinus montana* | 8 | Moder | 48 | 41 | 11 | -388 | -65 | -13 |

[1]Walthert et al. (2003) [2]Etzold et al., (2014), [3]Von Arx et al. (2013), [4]Krause et al., (2013)





**Table 3** Soil properties as well as carbon stocks and fluxes in 0-20, 20-60, and 60-100 cm depth of the study sites for the bulk and water-extractable organic carbon (WEOC).

| Location | Depth interval (m) | pH[1] | CEC[1] (mmolc/kg) | Fe_exchangeable (mmolc/kg) | Al_exchangeable (mmolc/kg) | Sand content (%) | Silt content (%) | Clay content (%) | Carbon stock kgC/m² | Average turnover bulk (y) | Average turnover WEOC (y) |
|---|---|---|---|---|---|---|---|---|---|---|---|
| Othmarsingen[1] | 0.0-0.2 | 4.4 | 62.2 | 0.15 | 42 | 46.8 | 35.5 | 17.6 | 4.84 | 162 | 35 |
| | 0.2-0.6 | 4.4 | 62.8 | 0.10 | 49 | 44.3 | 33.3 | 22.4 | 1.69 | 868 | 517 |
| | 0.6-0.8 | 4.9 | 99.5 | 0.06 | 41 | 46.7 | 28.4 | 25.0 | 0.28 | 3938 | - |
| Lausanne[1] | 0.0-0.2 | 4.5 | 60.8 | 0.13 | 43 | 49.2 | 32.6 | 18.2 | 3.24 | 298 | 77 |
| | 0.2-0.6 | 4.6 | 43.9 | 0 | 34 | 50.2 | 32.0 | 17.8 | 2.12 | 1197 | 586 |
| | 0.6-1.0 | 4.8 | 49.7 | 0 | 35 | 50.5 | 31.5 | 18.1 | 0.69 | 2242 | 1502[5] |
| Alptal[2,3,4] | 0.0-0.2 | 4.5 | 417 | - | 19 | 19.3 | 39.4 | 41.3 | 7.73 | 293 | 166 |
| | 0.2-0.6 | 4.7 | 340 | - | 14 | 4.90 | 47.0 | 48.1 | 7.24 | 2943 | 893[6] |
| | 0.6-1.0 | 4.7 | 340 | - | - | - | - | - | 6.54 | 5165 | - |
| Beatenberg[1] | Organic layer | 3.1 | 260.2 | 2.8 | 33 | - | - | - | 7.05 | 54 | - |
| | 0.0-0.2 | 4.0 | 35.6 | 1.7 | 18 | 84.9 | 12.4 | 2.7 | 3.65 | 1081 | 293 |
| | 0.2-0.6 | 4.1 | 23.1 | 0.40 | 17 | 83.2 | 12.3 | 4.6 | 4.10 | 1607 | 677 |
| Nationalpark[1] | 0.0-0.2 | 8.3 | 171.8 | 0.1 | 0.0 | 47.5 | 34.8 | 17.7 | 3.23 | 159 | 92 |
| | 0.2-0.6 | 8.8 | 106.3 | 0.0 | 0.0 | 61.9 | 32.5 | 5.7 | 0.36 | 612 | 214 |
| | 0.6-0.8 | - | - | 0.0 | 0.0 | 60.6 | 33.6 | 5.9 | 0.08 | 983 | - |

[1]Walthert et al., 2002, Walthert et al., 2003.; Fe and Al content (mmolc/kg) determined by NH₄Cl extraction.
[2]Krause et al., 2013
[3]Diserens et al.,1992, CEC determined (mmeq/kg), hydrogen lead and zinc ions were not included, Aluminium content determined by Lakanen method. CEC values for 0.2-0.4 m were determined by Lakanen method.
For the 0.2-0.6 depth interval the CEC determined for 0.2-0.4 m was taken, and similarly for the depth interval 0.6-1.0 m the values for 0.6-0.8 m were taken in the case of Othmarsingen, Lausanne Beatenberg and Nationalpark. [4]Xu et al., 2009 [5]Depth to 0.8 m [6]Depth to 0.4 m



**Table 4** Pearson correlations for averaged depth intervals for the top soil (0–20 cm, n=5) and deep soil (20–60 cm, n=5). Significance denoted with $\cdot$, $*$, $**$ or $***$ for respectively p-values smaller than 0.1 (marginally significant) 0.05, 0.005 and 0.0005 (significant). Non-significant correlations are indicated by the superscript **ns**. SWP or soil water potential used are the median values at 15 cm for each of these 5 sites (Von Arx et al., 2013). Water-extractable carbon is abbreviated to WEOC. Results indicate that no single climatic or textural factor consistently co-varies with carbon stocks, or turnover time.

| Explaining variable | Stock$_{0-20\ cm}$ | Turnover time bulk$_{0-20\ cm}$ | Turnover time WEOC$_{0-20\ cm}$ | Stock$_{20-60\ cm}$ | Turnover time$_{20-60\ cm}$ | Fraction dynamic$_{0-20\ cm}$ |
|---|---|---|---|---|---|---|
| MAT | 0.17$^{ns}$ | -0.12$^{ns}$ | -0.36$^{ns}$ | 0.08$^{ns}$ | 0.04$^{ns}$ | -0.15$^{ns}$ |
| MAP | **0.96**$^{*}$ | 0.04$^{ns}$ | 0.29$^{ns}$ | **0.95**$^{*}$ | **0.97**$^{**}$ | **0.98**$^{*}$ |
| NPP | 0.2$^{ns}$ | 0.68$^{ns}$ | 0.38$^{ns}$ | 0.07$^{ns}$ | -0.05$^{ns}$ | -0.36$^{ns}$ |
| Sand | -0.66$^{ns}$ | 0.77$^{ns}$ | 0.53$^{ns}$ | -0.58$^{ns}$ | -0.65$^{ns}$ | **-0.98**$^{*}$ |
| Silt | 0.38$^{ns}$ | **-0.94**$^{*}$ | -0.79$^{ns}$ | 0.29$^{ns}$ | -0.40$^{ns}$ | 0.84$^{ns}$ |
| Clay | **0.81**$^{\cdot}$ | -0.57$^{ns}$ | -0.29$^{ns}$ | 0.74$^{ns}$ | 0.79$^{ns}$ | **0.99**$^{*}$ |
| CEC | -0.67$^{ns}$ | -0.68$^{ns}$ | -0.50$^{ns}$ | **-0.98**$^{*}$ | **-0.98**$^{***}$ | 0.16$^{ns}$ |
| pH | -0.74$^{ns}$ | -0.49$^{ns}$ | -0.28$^{ns}$ | -0.78$^{ns}$ | -0.75$^{ns}$ | 0.20$^{ns}$ |
| Fe | 0.24$^{ns}$ | -0.66$^{ns}$ | -0.81$^{ns}$ | -0.17$^{ns}$ | -0.15$^{ns}$ | -0.01$^{ns}$ |
| Al | 0.18$^{ns}$ | -0.62$^{ns}$ | -0.77$^{ns}$ | -0.09$^{ns}$ | -0.0$^{ns}$ | -0.13$^{ns}$ |
| SWP | 0.70$^{ns}$ | 0.64$^{ns}$ | 0.71$^{ns}$ | - | - | 0.82$^{ns}$ |
| Average Grain size | -0.25$^{ns}$ | **0.95**$^{*}$ | **0.81**$^{\cdot}$ | 0.01$^{ns}$ | -0.1$^{ns}$ | -0.76$^{ns}$ |



**Figures**

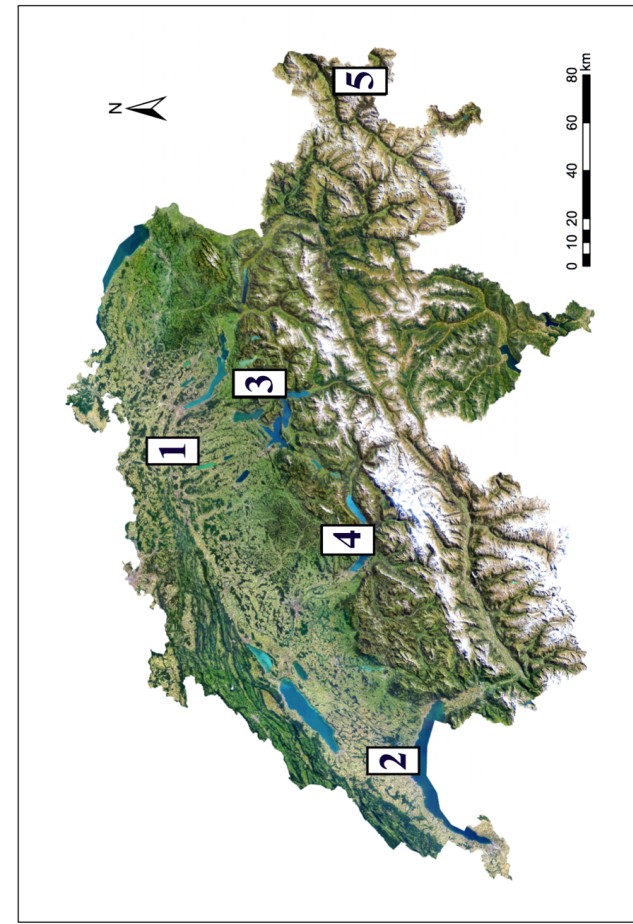

**Figure 1** Sample locations, all of which are part of the Long-term ecosystem research program (LWF) of the Swiss Federal Institute WSL, 1) Othmarsingen, 2) Lausanne, 3) Alptal, 4) Beatenberg and 5) Nationalpark Image made using 2016 swisstopo (JD100042).



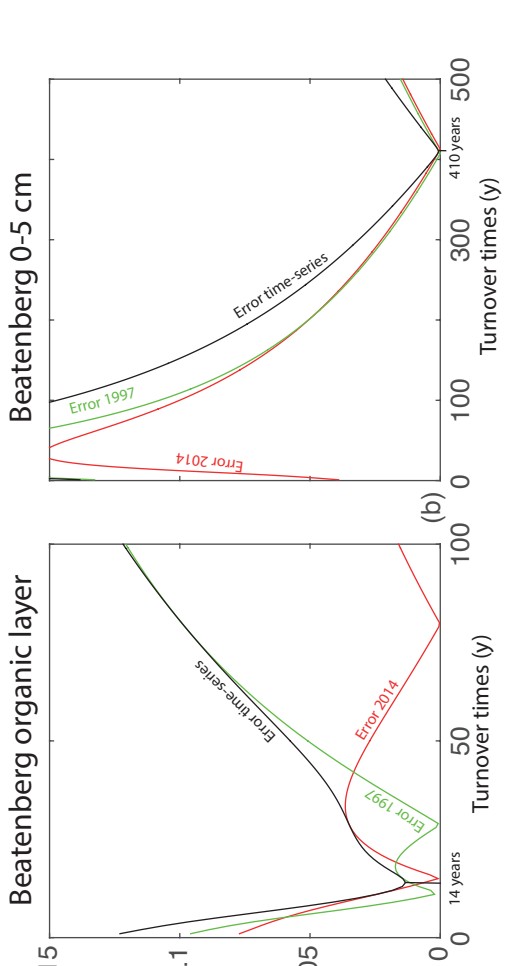

**Figure 2** Numerical optimization of least mean-square error reduction, showing and the reduction of error spread for two soil depths. For the Beatenberg organic layer (a) the individual time points yield two solutions are almost equally likely, but combined the time-points reveal the likeliest option. For the (b) 0-5 cm layer the single time points only have a single likely solution.



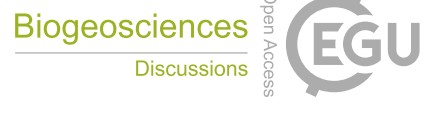

**Figure 3** (a) Time-series soil carbon turnover time in years (y) as determined by numerical modelling for (b) sub-alpine site Beatenberg. The bulk turnover in the organic layer is rapid (14 years), followed by the turnover of the water-extractable organic carbon (WEOC) (191 years) and the bulk turnover of the soil (410 years). Photo soil profile courtesy of Marco Walser, WSL.





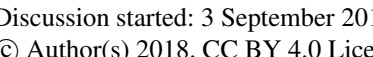

**Figure 4** (a–e) Changes in radiocarbon signature of both bulk soil and WEOC over two decades at four sites on a climatic gradient. For Alptal (c) only the 2014 time-point was available. For the warmer locations (Luvisol, Cambisol MAT 9.2-7.6 °C), depletion in bomb-derived radiocarbon occurs in the first five centimeters soil in 2014 as compared to 1995-8. The colder Beatenberg site (Podzol, MAT 4.7 °C) is marked by a clear enrichment of $^{14}$C in the mineral soil in 2014 w.r.t. 1997. At the coldest site Nationalpark (Fluvisol, MAT 1.3 °C) almost all samples taken two decades after the initial sampling show an enrichment in radiocarbon signature. WEOC contains bomb-derived carbon in the topsoil in 2014 at all sites.



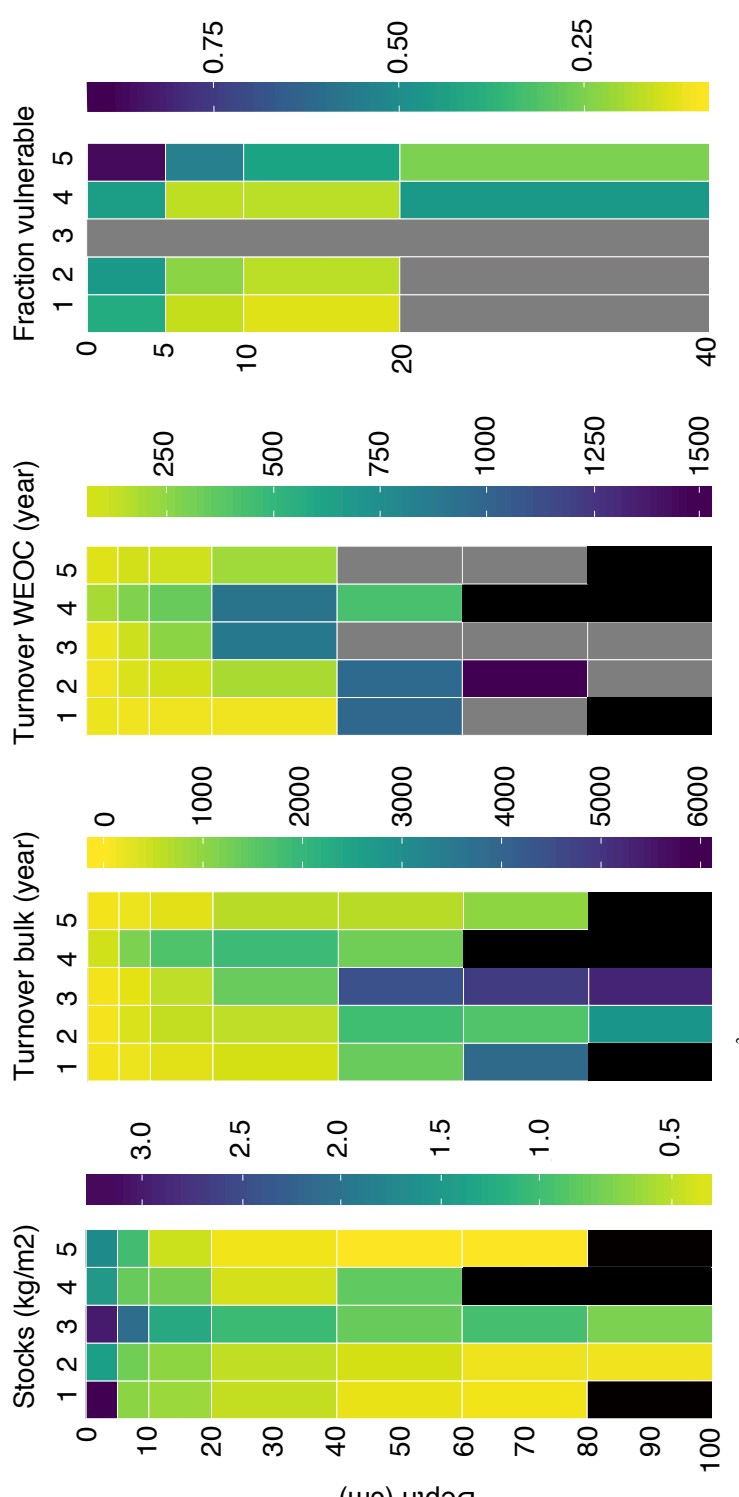

**Figure 5** Carbon (a) stocks in the mineral soil kgC/m$^2$, (b) turnover time bulk soil in years (c) turnover time water extractable organic carbon soil in years and (d) fraction vulnerable pool in 5 cm intervals. Locations are ordered from the warmest to coldest sites i.e. (1) Othmarsingen, (2) Lausanne, (3) Alptal, (4) Beatenberg and (5) Nationalpark. Grey boxes indicate absence of material, black boxes indicate the occurence of the C-horizon (poorly consolidated bedrock-derived stony material or bedrock itself).





**Figure 6** Modeled turnover times (y) of single profiles sampled down to the bedrock between 1995 and 1998. $\Delta^{14}$C published in Van der Voort et al. (2016). Results indicate presence of petrogenic (bedrock-derived) carbon as modeled turnover time exceeds soil formation since the end of last ice age (10,000 years) in Lausanne (>100 cm, Cambisol), Alptal (80-100 cm, Gleysol).