# Peer review of "Dynamics of deep soil carbon – insights from 14C time-series across a 1 2 climatic gradient"

_Biogeosciences, 2018_

## Referee Comment (RC1) · J. Balesdent (Referee) · 26 Sep 2018

Understanding the dynamics of carbon in deep soil layers is an important issue, and this study uses an excellent sequence and provides a rare dataset: soil 14C measurement at two dates using archived samples brings a precious information of C dynamics. One of the interesting results is the demonstration of the occurrence of rock-derived carbon. Another concerns the age of water extractable carbon. The analytical methods are high standard and highly relevant. I therefore consider it is worth publishing the data in Biogeochemistry. Unfortunately, there are major concerns that need revision. The most important is that the mathematical and numerical interpretations look

inappropriate, and this leads the authors to give conclusions that are in contrast with what the data show, whereas some unprecedented results could be derived. I finally suggest two alternative solutions: either the authors drop the modelling part and make a semi-quantitative interpretation of the data, either they use another model. I also noticed miscellaneous improvements to be done. The discussion should be updated according to these major points. The title and summary are nevertheless appropriate.

1. The chosen model is unlikely to simulate observed data.

Most of samples below 10 cm show an increase in $\Delta14C$ between 1990's and 2010's, by several 10‰ (Figure 3), and even some above 10 cm do. As seen in the FIGURE below, which was built for this review, the 14C content of well mixed compartments directly fed from atmospheric C has DECREASED with time since the 1990's (or increased by less than 4‰ for slow pools). The sum of two parallel pools cannot have a $\Delta14C$ increased between 1995 and 2014.

FIGURE: Simulated $\Delta14C$ of a well-mixed compartment under steady state as a function of compartment turnover rate, for two dates of sampling.

I finally understood (from 14C data in Figure 3 and turnover time data in Table S5) that the the "mosty reliable' kWSOC value is more or less the arithmetic mean of two kWSOC values, one calculated in the 1990's and the other in the 2010's. The authors must invoke other processes to explain an increasing $\Delta14C$. These processes may act together and interact:

- Transit of carbon in another horizons or pool before entering the observed layer. This might be associated with either bioturbation or DOC production from an above layer, movement, and insolubilization. The data tend to indicate that carbon movement is a significant cause of the increase in $\Delta14C$ across the sequence.

- non-steady state, e.g. increased bioturbation due to warming, change in NPP and/or decay rates.

To me, the fact that the ∆14C of WSOC of all samples (except Othmarsingen 0-5 cm and Lausanne 0-5 cm) inceased is a proof that WSOC is a by-product of SOM aged several 10th of years (usual age of OH horizons), and not directly fed by vegetation decomposition. This would be a bright finding and merit appropriate modelling.

2. Consistency in model implementation (to be confirmed).

I tried to calculate by myself turnover time values, based on 14C data in Figure 3 and turnover time data in Table S5, and didn't find the author's results. This may arise from the fact that the basic differential equations of the model (equation 5 = SI.7) looks false, or at least do not correspond to authors' hypotheses. Equation SI.7 states:

$F(t) = k \cdot Fatm(t) + m1 \cdot F(t - 1).(1 - \lambda - k1 ) + m2 \cdot F(t-1).(1 - \lambda - k2)$

This equation indicates that the flux of 14C leaving the system (out of desintegration) is:

$(m1.k1 + m2.k2).F(t -1)$, i.e., $k.F(t)$

Since the corresponding flux of carbon is $k = m1.k1 + m2.k2$, this equation says that the 14C activity of carbon leaving the system is $F(t - 1)$. So the equation would IMPLICITELY considers that the activity of the flux out is the same as that of the compartment itself. This is typically the assumption of a so-called 'well mixed' compartment, and is not the case of a system with two compartments. It would only accept the solution $k1 = k2$. Making this implicit assumption is a current mistake or at least a source of disagreement in isotope geochemisty. As a consequence, I guess that the authors have calculated a mean turnover time corresponding to a single compartment for bulk carbon, and an independent specific turnover time of WSOC. The error might be linked with my point 3 below. See a proposal for the correct equation as an appendix of this review. The authors are invited to check how eq SI.7 was implemented and how the couple ($k2$ , $m1$) was inferred from bulk F14C.

3. Mathematical (and semantic) misuse of "turnover time'.

Let us call the turnover time of carbon in the compartment T = 1/k Mathematically, the carbon input to the system is m1/T1 + m2/T2. The size of the compartment is m1 + m2. So, the turnover time, which is the ratio of pool size to the input, is:

T = (m1 + m2)/(m1/T1 + m2/T2)

In Table SI.5, which presents the main result, i.e. the values of turnover time, the authors calculated the bulk turnover time as:

T = (m1.T1 + m2.T2)/(m1 + m2), which is wrong.

What authors call "turnover time" is in fact the MEAN AGE of carbon, which is different of the mean turnover time in non-well mixed compartments. The error in not only semantic because it possibly have interfered in model and 14C equation (point 2). Sierra et al. (2016), whom you cite lines 161-162, recommends the use of "age", not "turnover time" for this variable. See also Manzoni et al.(2009).

4. Data availability.

The authors must provide in SI a table including the primary data, i.e., $\Delta$14C, C stock by horizon, WEOC stocks. Reference that were used to estimate atmospheric $\Delta$14C (post bomb and pre-bomb) should be indicated (e.g. Reimer , Hua etc.)

5. Hypothesis on WSOC as the labile pool.

Line 180-182 and 190-191: A major (if not the major) assumption of the model is that the dynamic pools has the same decay rate as that of WEOC. The 'dynamic' pools contains as much as 88% of soil C (on the average 34%), whereas WEOC only a few %. Assigning the constant k of WEOC to the dynamic pool is therefore a surprising and very heavy hypothesis. (see also point 1.)

Alternatively, the study may have targetted the study of WSOC dynamics for itself, e.g., considered that both WSOC and bulk C are heterogenous pools, each with a labile and a more stable component, but in varied proportion. Many other models use particulate

organic matter (i.e. either sand-size primary organic particles or light OM, which has been described as having a good fit with labile carbon.

6. Conclusions on correlation with MAP.

Projecting conclusions on the effect of MAP on the basis of a "wet" sequence, i.e., where the water deficit is probably low if not nil, may look brash. The driest site is 800 mm, but with a MAT 1.3°C and probably a small PET. Furthermore (Lines 360-361), authors state that 'The only climate-related driver which appears to be significant is precipitation' whereas the r2 coefficient between MAP and turnover 0-20 cm is 0.04! I would recommend here to cite Carvalhais et al. (2013) and Mathieu et al. (2015), who highlighted the role of precipitation in SOM stabilization or ecosystem carbon turnover. I sfinally uggest to moderate the conclusions, but maybe discuss the role of precipitatioon on DOC movement (see point 1).

7. Presentation of model and equations.

The presentation of both the model and the optimization process is obscure throughout the text and should be more precise, in either text or SI. In the cases with four radiocarbon dates (2 sampling dates x two fractions), the optimization of three dynamic parameters is not a formal solution, but a best fit. The type of adjustment (least squares ?) and a criterion of the fit (e.g., RMSE) should be indicated. Harmonize the name of variables throughout the text and SI. For consistency with SI, please use m insteaf of F in eqn (3), (4) and (4); and possibly F instead of R. Also use the same character k in SI and main text. Harmonize M (Figure S2) and m, etc. How were single points managed ? (Line 194-195. " Due to limited availability of archived samples, there are only single time points available for some samples as indicated in Fig. 4.")

8. Miscellaneous.

lInes 51-52 note the pioneer studies by Jenkinson et al (1992) on long-term experiments. The models by Braakkeke et al. (2014 ) also simulates 14C profiles in rather

similar podzols, using WSOC as well, and may receive more attention in the discussion section. Also note (e.g. Line 34) the conclusions of Mathieu et al. (2015) concerning soil versus climate drivers of 14C, and (lines 39-40) the recent paper by Balesdent et al. (2018), which improved the understanding of the significance of deep soil C to the global C cycle.

Move lines 126-128 (WEOC) to the end of 2.1. (WEOC extraction). Note that extraction with Na 0.86 M is not exactly Water extraction, since it moves some exchangeable calcium, disperses clays and therefore moves sorbed organic compounds that would not have been mobilized by water.

Line 252 ' Deeper soil bulk stock and turnover positively...' and table S5: avoid "turnover " alone standing for "turnover time" in such sentences, because the common sense of turnover is turnover rate, i.e., the inverse of turnover time. This may lead to a reverse understanding of correlations.

Line 262. Balesdent et al. (2018) reported that 21% of world subsoil C (30-100 cm) is less than 50 years old.

The amount of WEOC (while not used in the modelling experiment) would be welcome.

Surprisingly, the section of Material and methods indicates that NPP and its components were measured, which is a rare information in SOM studies. As a result, authors have an indicator of the true turnover time of soil C, i.e. the ratio of Soil C stock to C input is known, that they do not use.

Figure 4 contains the main primary result of the study. Polices Should be enlarged. The square signs for Aptal WEOC 1997 are misleading. Table S5 is the main final result and should take place in the main document.

Note that the bi-exponential age distribution is factually the age distribution of C in current "four pools" models such as RothC (or Century). All coupling of these models with radiocarbon more or less managed bi-exponential age distribution and 14C; e.g.,

Jenkinson et al. (1992).

9. Appendix

The differential equation should consider F1 and F2 the 14C fraction in pools 1 and 2, respectively, as illustrated in your Fig S1.

Input flux to pool1 is k1.m1; input flux to pool2 is k2.m2

F1(t) =k1.Fatm(t) + (1 - k1 - $\lambda$).F1(t - 1)

F2(t) =k2.Fatm(t) + (1 - k2 - $\lambda$).F2(t - 1)

which give: F(t) = m1F1(t) + m2.F2(t) = k.Fatm(t) + m1.(1 - k1 - $\lambda$).F1(t - 1) + m1.(1 − k2 - $\lambda$).F2(t - 1)

And needs numerical resolution of F1 and F2.

10. Cited references Balesdent J., Basile-Doelsch I, Chadoeuf J., Cornu S., Derrien D. Fekiacova Z., Hatté C. Atmosphere-soil carbon transfer as a function of soil depth. Nature, 559, 599–602. (2018) doi.org/10.1038/s41586-018-0328-3

Jenkinson D.S., D.D. Harkness, E.D. Vance, D.E. Adams and A.F. Harrison. Calculating net primary production and annual input of organic matter to soil from the amount and radiocarbon content of soil organic matter. Soil Biol. Biochem. 24(4):295-308 (1992)

Manzoni, S., Katul, G. G. & Porporato, A. Analysis of soil carbon transit times and age distributions using network theories. J. Geophys. Res. 114, G04025 (2009)

Mathieu J., Hatté C., Parent E., Balesdent J. Deep soil carbon dynamics are driven more by soil type than by climate: a worldwide meta-analysis of radiocarbon profiles. Global Change Biology 21, 4278-4292. (2015) doi:10.1111/gcb.13012.

11. Figure.

Simulated $\Delta$14C of a well-mixed compartment under steady state as a function of compartment turnover rate, for two dates of sampling. Compartment has a single

exponential distribution of ages; system start 8050 BP; atmospheric ∆14C after Reimer et al. (2009) and Hua et al. (2013); Northern hemisphere zone N2; May-August.

[Figure]

Fig. 1. Simulated Δ14C of a well-mixed compartment under steady state as a function of compartment turnover rate, for two dates of sampling. Compartment has a single exponential distribution of ages;

---

## Referee Comment (RC2) · Anonymous Referee #2 · 28 Sep 2018

This study aims at investigating the dynamics of carbon as a function of soil depth in five sites of the Swiss Alps. To reach this goal the authors realised 14C measurements on samples collected in late 90's and in 2014. Soils were sampled at different depths and a water extractable fraction was extracted. The authors derived C turnover rates from 14C data using a two-pool model. They identify a substantial fraction of fast-cycling C at depth and further investigate potential edaphic and climatic drivers of turnover.

The data gathered in this study are of great interest, but at this stage, the manuscript suffers from too severe limitations to be published.

In particular, the authors should decide what is precisely their objective: do they want

to provide insights on deep C cycling or to offer a new method to compute turn-over time using 14C data? I would suspect the readers of Biogeosciences to be really interested in the first option, as there are only a limited number of studies on this topic (as claimed in l 276 of the discussion). Nevertheless, the data on C turn-over along the soil profile are mainly presented as supplementary, while there is a strong focus on methodological aspects in the main text.

The discussion should also be improved. Too many repetitions of the results in 4.1.1 and 4.1.2; 4.1.3 repeats some facts of 4.1.2. 4.2: I could not find clear information in the materials and methods section about how the data supporting this section were collected. The introduction/rational should refer to the needs of information on petro-genic C. 4.3: you could condense your message as you expose the same arguments for bulk C and WEOC.

Some references to recent publications on deep C dynamics are lacking (i.e. He et al., 2016; Mathieu et al 2016; Balesdent et al 2018) while they could improve the discussion.

I finally encourage the authors to carefully examine the relevance - and the quality - of their illustrations (see some comments below). A better focus of both the text and illustrations would guarantee a better understanding of the message the authors could deliver from the very exceptional dataset they collected.

Some additional comments

Could you indicate what is "Rsample,t" in Eq 1 and 2. The model is based on the assumption that k1 is the turn-over of the WEOC pool. However, how do you justify that m1 is not the size of the WEOC pool (please provide the C content of the WEOC in your MS). ?

Clarify what do you mean by deep, and provide numerical value when you refer to depth in the text – currently you sometimes use it indifferently to refer to 30 cm or 80

cm, while the data strongly differ between both depths.

Some Figures and Tables are offered to the readers while they are not utilised in the text: remove them (one example is Fig 3 - PS the information on the back curve is missing in the legend)

I do not understand Figure 2. How do you compute turn-over time using one individual time point?

I suggest to remove Figure 5 as it is not precise – keep it for oral presentations - (what is vulnerable C?) and to provide Tables with exact numerical data in the main text.

Please provide the C content in for the samples measured for 14C. (Table 3 only show 3 different depths, while the data is available according to Fig 5)

You provide twice the particle size distribution (Tab 2 and 3).

Some of your interpretations rely on soil waterlogging while this information is not clearly available (when you first mention waterlogged soil line224, the reader has not idea of which sites are concerned). In addition, I would not conclude that waterlogging is a driver of turnover by looking at the non-averaged values in Table S5.

Why are the radiocarbon signatures of WEOC different between waterlogged and non-waterlogged soils in 3.1, while calculated turnover rates are not?

Change your title: your gradient is not only a climatic one but a geologic one as well, with strong implication on C cycling.

Figure 6: the colour code is not the same than in other figures.

I do not understand Table S1: how do you compute single resolved 14C data?

Fig S2: what stands at -20cm depth?

Table S5: figures are not aligned in the table what makes the reading a bit tricky. The caption is not in the same order than the columns. The title of the 5th column is not

clear (=> proportion of labile pool would be better)
* * *

---

## Author Comment (AC1) · 8 Feb 2019

Response to Jerome Balesdent Understanding the dynamics of carbon in deep soil layers is an important issue, and this study uses an excellent sequence and provides a rare dataset: soil 14C measure- ment at two dates using archived samples brings a precious information of C dynamics. One of the interesting results is the demon- stration of the occurrence of rock-derived carbon. Another concerns the age of water extractable carbon. The analytical meth- ods are high standard and highly relevant. I therefore consider it is worth publishing the data in Biogeochemistry. Unfortunately, there are major concerns that need revi- sion. The most important is that the mathematical and numerical interpretations look inappropriate, and this leads the authors to give conclusions that are in contrast with what the data show, whereas some unprecedented results could be derived. I finally suggest two alternative solutions: either the authors drop the modelling part and make a semi-quantitative interpretation of the data, either they use another model. I also noticed miscellaneous improvements to be done. The discussion should be updated according to these major points. The title and summary are nevertheless appropriate. Dear Prof. Balesdent,

–> Thank you very much for your positive feedback and thorough review. We very much appreciate that you value the importance of the data for the wider Biogeosciences community. Your comments about the turnover time modelling are also very insightful and the issues have subsequently been addressed. There was indeed a semantic issue which caused problems, so we incorporated all of your feedback. We realized that most of the modelling was explained in the code in the SI, and that therefore the text in the main text was absolutely inadequate in order to explain our calculations. Consequently, paper and especially the discussion was updated according to these major points. As you indicated, the title and summary remained appropriate.

We want to thank you again for your helpful review, which has further improved this paper. Please find detailed replies below. 1. The chosen model is unlikely to simulate observed data. Most of samples below 10 cm show an increase in $\Delta 14C$ between 1990's and 2010's, by several 10‰ (Figure 3), and even some above 10 cm do. As seen in the FIGURE below, which was built for this review, the 14C content of well mixed compartments directly fed from atmospheric C has DECREASED with time since the 1990's (or in- creased by less than 4‰ for slow pools). The sum of two parallel pools cannot have a $\Delta 14C$ increased between 1995 and 2014. FIGURE: Simulated $\Delta 14C$ of a well-mixed compartment under steady state as a func- tion of compartment turnover rate, for two dates of sampling. Thank you for these comments. Indeed, rapidly turning-over compartments have decreased in 14C in the last two decades, whilst the slower compartments have increased in 14C signature (e.g. Figure 3a) (as you also indicate in your comments below). As you indicated in your supplemental, there was indeed a semantic issue with the turnover time definition when estimating the size of the two respective pools, which has now been adapted. Our apologies for the confusion.

I finally understood (from 14C data in Figure 3 and turnover time data in Table S5) that the the "mosty reliable' kWSOC value is more or less the arithmetic mean of two kWSOC values, one calculated in the 1990's and the other in the 2010's. The authors must invoke other processes to explain an increasing $\Delta$14C. These processes may act together and interact: - Transit of carbon in another horizons or pool before entering the observed layer. This might be associated with either bioturbation or DOC production from an above layer, movement, and insolubilization. The data tend to indicate that carbon movement is a significant cause of the increase in $\Delta$14C across the sequence. - non-steady state, e.g. increased bioturbation due to warming, change in NPP and/or decay rates. To me, the fact that the $\Delta$14C of WSOC of all samples (except Othmarsingen 0-5 cm and Lausanne 0-5 cm) inceased is a proof that WSOC is a by-product of SOM aged several 10th of years (usual age of OH horizons), and not directly fed by vegetation decomposition. This would be a bright finding and merit appropriate modelling. Thank you for these comments as well. We did not sufficiently explain how we estimate the turnover of the WEOC or'labile' pool using the 14C time-series. We have addressed this now in the method section, by detailing the different steps and the error calculation (Equations 1-4). In short, we do not take the arithmetic mean, but rather use the standard equations (e.g. Herold et al., 2014; Torn et al., 2009) to find the likeliest turnover time considering both datapoints of the time series. Instead of the usual excel-based method, we do this in MatLab because it is automated and more repeatable. The solution which has the lowest calculated residual square root mean error (RSME) is automatically chosen, as opposed to a manual iteration.

–> Thank you for highlighting the importance of potentially DOC-driven transport of young(er) carbon through the deep soil, we have included this in our discussion. We have also now included your suggestion in Section 4.1.3 to highlight that WEOC is likely not fed by vegetation decomposition but rather is derived from several decades-old SOM.

2. Consistency in model implementation (to be confirmed). I tried to calculate by myself turnover time values, based on 14C data in Figure 3 and turnover time data in Table S5, and didn't find the author's results. This may arise from the fact that the basic differential equations of the model (equation 5 = SI.7) looks false, or at least do not correspond to authors' hypotheses. Equation SI.7 states: F(t) = kÅůFatm(t) + m1ÅůF(t - 1).(1 - $\lambda$ - k1 ) + m2ÅůF(t-1).(1 - $\lambda$ - k2) This equation indicates that the flux of 14C leaving the system (out of desintegration) is: (m1.k1 + m2.k2).F(t -1), i.e., k.F(t) Since the corresponding flux of carbon is k = m1.k1 + m2.k2, this equation says that the 14C activity of carbon leaving the system is F(t – 1). So the equation would IM- PLICITELY considers that the activity of the flux out is the same as that of the compart- ment itself. This is typically the assumption of a so-called 'well mixed' compartment, and is not the case of a system with two compartments. It would only accept the solu- tion k1 = k2. Making this implicit assumption is a current mistake or at least a source of disagreement in isotope geochemisty. As a consequence, I guess that the authors have calculated a mean turnover time corresponding to a single compartment for bulk carbon, and an independent specific turnover time of WSOC. The error might be linked with my point 3 below. See a proposal for the correct equation as an appendix of this review. The authors are invited to check how eq SI.7 was implemented and how the couple (k2 , m1) was inferred from bulk F14C.

–> Thank you, there are two main things raised in this comment: A. Modelling Structure Indeed, we have calculated a mean turnover time corresponding to a single compart- ment for bulk carbon, and one independent specific turnover time of WSOC. We have clarified this in the text. B. Model consistency Thank you for your suggestions and example for Figure SI.7, we have implemented all of your suggestions (Eq. 6). More details can be found below. We would like to clarify that we merely transformed the usual excel file people use to find turnover time to MatLab-driven optimization, because it saves time, is repeatable, unbiased and error can be quantified. We have now also quantified all our errors (See SI). Furthermore, the code can easily be used as well for longer time-series (i.e. > 2 timepoints). We benchmarked our results to the Excel-method, and the results agree.

3. Mathematical (and semantic) misuse of "turnover time'. Let us call the turnover time of carbon in the compartment $T = 1/k$ Mathematically, the carbon input to the system is $m1/T1 + m2/T2$. The size of the compartment is $m1 + m2$. So, the turnover time, which is the ratio of pool size to the input, is: $T = (m1 + m2)/(m1/T1 + m2/T2)$ In Table SI.5, which presents the main result, i.e. the values of turnover time, the authors calculated the bulk turnover time as: $T = (m1.T1 + m2.T2)/(m1 + m2)$, which is wrong. What authors call "turnover time" is in fact the MEAN AGE of carbon, which is different of the mean turnover time in non-well mixed compartments. The error in not only semantic because it possibly have interfered in model and 14C equation (point 2). Sierra et al. (2016), whom you cite lines 161-162, recommends the use of "age", not "turnover time" for this variable. See also Manzoni et al.(2009).

–> Indeed, there was a (semantic) inconsistency regarding turnover time between the Main text and the SI, which we have now addressed and corrected. We also implemented your equation. We have the 14C-determined 'turnover time' for the bulk soil, whilst stating that we assume a steady state. We have also clarified our definition in the text, following Manzoni et al. (2009) as well.

4. Data availability. The authors must provide in SI a table including the primary data, i.e., $\Delta$14C, C stock by horizon, WEOC stocks. Reference that were used to estimate atmospheric $\Delta$14C (post bomb and pre-bomb) should be indicated (e.g. Reimer , Hua etc.)

–> We have included an excel file with all the raw data regarding $\Delta$14C and stocks the WEOC material. The WEOC concentration is low (< 1 %) and can be found in SI Table 4. We had indicated the provenance of our pre- and post-bomb data already in the method section, but we have now further clarified it.

5. Hypothesis on WSOC as the labile pool. Line 180-182 and 190-191: A major (if not the major) assumption of the model is that the dynamic pools has the same decay rate as that of WEOC. The 'dynamic' pools contains as much as 88% of soil C (on the average 34%), whereas WEOC only a few %. Assigning the constant k of WEOC to the dynamic pool is therefore a surprising and very heavy hypothesis. (see also point 1.) Alternatively, the study may have targetted the study of WSOC dynamics for itself, e.g., considered that both WSOC and bulk C are heterogenous pools, each with a labile and a more stable component, but in varied proportion. Many other models use particulate organic matter (i.e. either sand-size primary organic particles or light OM, which has been described as having a good fit with labile carbon

–> Yes indeed, it was our assumption is we assumed the measured WEOC could be representative of the dynamic pool. There are studies that hypothesize WEOC could be indicative of a larger dynamic pool (Baisden and Parfitt, 2007; Koarashi et al., 2012). But indeed, this is a heavy assumption. We have therefore decreased the importance of the two-pool model in the paper, and highlighted this assumption. Indeed, both the WEOC and bulk themselves can heterogeneous pools, hence we also looked at biomarkers in another study (e.g. Van der Voort et al., 2017, Diverse Soil Carbon Dynamics Expressed at the Molecular Level, GRL). Looking at other fraction would be a worthwhile topic for future work.

6. Conclusions on correlation with MAP. Projecting conclusions on the effect of MAP on the basis of a "wet" sequence, i.e., where the water deficit is probably low if not nil, may look brash. The driest site is 800 mm, but with a MAT 1.3C and probably a small PET. Furthermore (Lines 360-361), authors state that 'The only climate-related driver which appears to be significant is precipitation' whereas the r2 coefficient between MAP and turnover 0-20 cm is 0.04! I would recommend here to cite Carvalhais et al. (2013) and Mathieu et al. (2015), who highlighted the role of precipitation in SOM stabilization or ecosystem carbon turnover. I sfinally uggest to mederate the conclusions, but maybe discuss the role of precipitation on DOC movement (see point 1).

–> Thank you for these insights. As suggested, we have highlighted the role of precipitation as SOM stabilizer and interaction with DOC movement, and tempered our statements about precipitation. Indeed, Switzerland is a wet country! Your own 2018 paper also could show the important role of evapotranspiration but we unfortunately do not have this data. Also, we adapted the phrasing of line 360-61, the role of precipitation is pronounced for the deep soil. 7. Presentation of model and equations. The presentation of both the model and the optimization process is obscure throughout the text and should be more precise, in either text or SI. In the cases with four radio- carbon dates (2 sampling dates x two fractions), the optimization of three dynamic parameters is not a formal solution, but a best fit.

–> Indeed, we have now mentioned this specifically in the text.

The type of adjustment (least squares ?) and a criterion of the fit (e.g., RMSE) should be indicated.

–> This has been included in the main text instead of the SI, we use RSME. Harmonize the name of variables throughout the text and SI. For consistency with SI, please use m insteaf of F in eqn (3), (4) and (4); and possibly F instead of R. Also use the same character k in SI and main text. Harmonize M (Figure S2) and m, etc. Thank you, this has now been adjusted. How were single points managed ? (Line 194-195. " Due to limited availability of archived samples, there are only single time points available for some samples as indicated in Fig. 4.")

–> This has been clarified in the main text, we solve the standard radiocarbon decay equations (e.g. Torn et al., 2009, ). This is done more traditionally in Excel, we did the same using a Matlab optimization.

8. Miscellaneous. lines 51-52 note the pioneer studies by Jenkinson et al (1992) on long-term experi- ments. The models by Braakkeke et al. (2014 ) also simulates

14C profiles in rather similar podzols, using WSOC as well, and may receive more attention in the discussion section. Also note (e.g. Line 34) the conclusions of Mathieu et al. (2015) concerning soil versus climate drivers of 14C, and (lines 39-40) the recent paper by Balesdent et al. (2018), which improved the understanding of the significance of deep soil C to the global C cycle.

–> Thank you, I have incorporated these literature suggestions. I had already cited Braakhekke et al.

Move lines 126-128 (WEOC) to the end of 2.1. (WEOC extraction). Note that extraction with Na 0.86 M is not exactly Water extraction, since it moves some exchangeable calcium, disperses clays and therefore moves sorbed organic compounds that would not have been mobilized by water.

–> Indeed, we followed Hagedorn et al., 2014 when preparing the extraction, and have this stated this clearly in the method section.

Line 252 ' Deeper soil bulk stock and turnover positively...' and table S5: avoid "turnover " alone standing for "turnover time" in such sentences, because the common sense of turnover is turnover rate, i.e., the inverse of turnover time. This may lead to a reverse understanding of correlations.

–> Indeed! We adapted this now.

Line 262. Balesdent et al. (2018) reported that 21% of world subsoil C (30-100 cm) is less than 50 years old. We have included this The amount of WEOC (while not used in the modelling experiment) would be welcome.

–> We have included this in the SI Table 4. Concentrations are low (< 1%)

Surprisingly, the section of Material and methods indicates that NPP and its components were measured, which is a rare information in SOM studies. As a result, authors have an indicator of the true turnover time of soil C, i.e. the ratio of Soil C stock to C input is known, that they do not use.

–> Indeed, there is NPP data, but we were recommended by the field experts that although it was representative for the tree vegetation, we had better not use it for estimating soil flux, as there would be too many assumptions to be considered. We did include the data, so others are free to use it.

Figure 4 contains the main primary result of the study. Polices Should be enlarged. The square signs for Aptal WEOC 1997 are misleading. Table S5 is the main final result and should take place in the main document.

–> We have adapted this figure slightly. Following your critique about the assumption of using WEOC as a dynamic pool we reduced the importance of the fraction modelling in the paper, so we opted to keep it in the SI.

–> Note that the bi-exponential age distribution is factually the age distribution of C in current "four pools" models such as RothC (or Century). All coupling of these models with radiocarbon more or less managed bi-exponential age distribution and 14C; e.g., Jenkinson et al. (1992).

–> Yes, we are familiar with Century (RothC), but feel applying them would be beyond the scope of this paper.

9. Appendix The differential equation should consider F1 and F2 the 14C fraction in pools 1 and 2, respectively, as illustrated in your Fig S1. Input flux to pool1 is k1.m1; input flux to pool2 is k2.m2 F1(t) =k1.Fatm(t) + (1 - k1 - $\lambda$).F1(t - 1) F2(t) =k2.Fatm(t) + (1 - k2 - $\lambda$).F2(t - 1) which give: F(t) = m1F1(t) + m2.F2(t) = k.Fatm(t) + m1.(1 - k1 - $\lambda$).F1(t - 1) + m1.(1 − k2 - $\lambda$).F2(t - 1) And needs numerical resolution of F1 and F2. –> Thank you, we implemented this. 10. Cited references Balesdent J., Basile-Doelsch I, Chadoeuf J., Cornu S., Derrien D. Fekiacova Z., Hatté C. Atmosphere-soil carbon transfer as a function of soil depth. Nature, 559, 599–602. (2018) doi.org/10.1038/s41586-018-0328-3 Jenkinson D.S., D.D. Harkness, E.D. Vance, D.E. Adams and A.F. Harrison. Calculating net primary production and annual input of organic matter to soil from the amount and radiocarbon content of soil organic matter. Soil Biol. Biochem. 24(4):295-308 (1992) Manzoni, S., Katul, G. G. & Porporato, A. Analysis of soil carbon transit times and age distributions using network theories. J. Geophys. Res. 114, G04025 (2009) Mathieu J., Hatté C., Parent E., Balesdent J. Deep soil carbon dynamics are driven more by soil type than by climate: a worldwide meta-analysis of radiocarbon profiles. Global Change Biology 21, 4278-4292. (2015) doi:10.1111/gcb.13012. –> Thank you, we implemented these papers 11. Figure. Simulated ∆14C of a well-mixed compartment under steady state as a function of compartment turnover rate, for two dates of sampling. Compartment has a single C7 exponential distribution of ages; system start 8050 BP; atmospheric ∆14C after Reimer et al. (2009) and Hua et al. (2013); Northern hemisphere zone N2; May-August. Interactive comment on Biogeosciences Discuss., https://doi.org/10.5194/bg-2018-361, 2018.

Please also note the supplement to this comment:
https://www.biogeosciences-discuss.net/bg-2018-361/bg-2018-361-AC1-supplement.zip
* * *
**Response to Jerome Balesdent**

Understanding the dynamics of carbon in deep soil layers is an important issue, and this study uses an excellent sequence and provides a rare dataset: soil 14C measure- ment at two dates using archived samples brings a precious information of C dynamics. One of the interesting results is the demonstration of the occurrence of rock-derived carbon. Another concerns the age of water extractable carbon. The analytical meth- ods are high standard and highly relevant. I therefore consider it is worth publishing the data in Biogeochemistry. Unfortunately, there are major concerns that need revi- sion. The most important is that the mathematical and numerical interpretations look inappropriate, and this leads the authors to give conclusions that are in contrast with what the data show, whereas some unprecedented results could be derived. I finally suggest two alternative solutions: either the authors drop the modelling part and make a semi-quantitative interpretation of the data, either they use another model. I also noticed miscellaneous improvements to be done. The discussion should be updated according to these major points. The title and summary are nevertheless appropriate.

Dear Prof. Balesdent,

Thank you very much for your positive feedback and thorough review. We very much appreciate that you value the importance of the data for the wider Biogeosciences community. Your comments about the turnover time modelling are also very insightful and the issues have subsequently been addressed. There was indeed a semantic issue which caused problems, so we incorporated all of your feedback. We realized that most of the modelling was explained in the code in the SI, and that therefore the text in the main text was absolutely inadequate in order to explain our calculations. Consequently, paper and especially the discussion was updated according to these major points. As you indicated, the title and summary remained appropriate.

We want to thank you again for your helpful review, which has further improved this paper. Please find detailed replies below.

1. The chosen model is unlikely to simulate observed data.

Most of samples below 10 cm show an increase in $\Delta14C$ between 1990's and 2010's, by several 10‰ (Figure 3), and even some above 10 cm do. As seen in the FIGURE below, which was built for this review, the 14C content of well mixed compartments directly fed from atmospheric C has DECREASED with time since the 1990's (or in- creased by less than 4‰ for slow pools). The sum of two parallel pools cannot have a $\Delta14C$ increased between 1995 and 2014.

FIGURE: Simulated $\Delta14C$ of a well-mixed compartment under steady state as a func- tion of compartment turnover rate, for two dates of sampling.

Thank you for these comments. Indeed, rapidly turning-over compartments have decreased in [14]C in the last two decades, whilst the slower compartments have increased in [14]C signature (e.g. Figure 3a) (as you also indicate in your comments below). As you indicated in your supplemental, there was indeed a semantic issue with the turnover time definition when estimating the size of the two respective pools, which has now been adapted. Our apologies for the confusion.

**Fig. 1.** Point by point replies
Interactive
comment

**Dynamics of deep soil carbon – insights from [14]C time-series across a**
**climatic gradient**
Tessa Sophia van der Voort[1], Utsav Mannu[1,†], Frank Hagedorn[2], Cameron McIntyre[1,3], Lorenz Walthert[2],
Patrick Schleppi[2], Negar Haghipour[1], Timothy Ian Eglinton[1]
[1]Institute of Geology, ETH Zürich, Sonneggstrasse 5, 8092 Zürich, Switzerland
[2]Forest soils and Biogeochemistry, Swiss Federal Research Institute WSL, Zürcherstrasse 111, 8903
Birmensdorf, Switzerland
[3]Department of Physics, Laboratory of Ion Beam Physics, ETH Zurich, Schaffmattstrasse 20, 9083 Zurich
[†]New address: Department of Earth and Climate Science, IISER Pune, Pune, India
*correspondence to:* Tessa Sophia van der Voort (tessa.vandervoort@erdw.ethz.ch)

**Abstract.** Quantitative constraints on soil organic matter (SOM) dynamics are essential for comprehensive
understanding of the terrestrial carbon cycle. Deep soil carbon is of particular interest, as it represents large
stocks and its turnover rates remain highly uncertain. In this study, SOM dynamics in both the top and deep soil
across a climatic (average temperature ~1-9 °C) gradient are determined using time-series (~20 years) [14]C data
from bulk soil and water-extractable organic carbon (WEOC). Analytical measurements reveal enrichment of
bomb-derived radiocarbon in the deep soil layers on the bulk level during the last two decades. The WEOC pool
is strongly enriched in bomb-derived carbon, indicating that it is a dynamic pool. Turnover time estimates of
both the bulk and WEOC pool show that the latter cycles up to a magnitude faster than the former. The presence
of bomb-derived carbon in the deep soil, as well as the rapidly turning WEOC pool across the climatic gradient
implies that there likely is a dynamic component of carbon in the deep soil. Precipitation and bedrock type
appear to exert a stronger influence on soil C turnover and stocks as compared to temperature.

**1  Introduction**
Within the broad societal challenges accompanying climate and land use change, a better understanding of the
drivers of turnover of carbon in the largest terrestrial reservoir of organic carbon, as constituted by soil organic
matter (SOM), is essential (Batjes, 1996; Davidson and Janssens, 2006; Doetterl et al., 2015; Prietzel et al.,
2016). Terrestrial carbon turnover remains one of the largest uncertainties in climate model predictions
(Carvalhais et al., 2014; He et al., 2016). At present, there is no consensus on the net effect that climate and land
use change will have on SOM stocks (Crowther et al., 2016; Gosheva et al., 2017; Melillo et al., 2002; Schimel
et al., 2001; Trumbore and Czimczik, 2008). Deep soil carbon is of particular interest because of its large stocks
(Jobbagy and Jackson, 2000; Balesdent et al., 2018; Rumpel and Kogel-Knabner, 2011) and perceived stability.
The stability is indicated by low [14]C content (Rethemeyer et al., 2005; Schrumpf et al., 2013; van der Voort et
al., 2016) and low microbial activity (Fierer et al., 2003). Despite its importance, deep soil carbon has been
sparsely studied and remains poorly understood (Angst et al., 2016; Mathieu et al., 2016; Rumpel and Kogel-
Knabner, 2011). The inherent complexity of SOM and the multitude of drivers controlling its stability further
impedes the understanding of this globally significant carbon pool (Schmidt et al., 2011). In this framework,
there is a particular interest in the portion of soil carbon that could be most vulnerable to change, especially in
colder climates (Crowther et al., 2016). Water-exactable organic carbon (WEOC) is seen as a dynamic and
potentially vulnerable carbon pool in the soil (Hagedorn et al., 2004; Lechleitner et al., 2016). Radiocarbon
([14]C) can be a powerful tool to determine the dynamics of carbon turnover over decadal to millennial timescales

**Fig. 2.** Paper included changes
* * *
[Figure]

**Fig. 3.** Paper tracked changes

---

## Author Comment (AC2) · 8 Feb 2019

Response to Reviewer #2

This study aims at investigating the dynamics of carbon as a function of soil depth in five sites of the Swiss Alps. To reach this goal the authors realised 14C measurements on samples collected in late 90's and in 2014. Soils were sampled at different depths and a water extractable fraction was extracted. The authors derived C turnover rates from 14C data using a two-pool model. They identify a substantial fraction of fast-cycling C at depth and further investigate potential edaphic and climatic drivers of turnover. The data gathered in this study are of great interest, but at this stage, the manuscript suffers from too severe limitations to be published.

–> Thank you very much for your positive feedback regarding the quality of the dataset and insights which can be gained from it. You indicated that the main limitation was the two-pool modelling, so we addressed this, details below. We have also addressed the other issues that have been raised. Thanks again for your helpful review, it helped further improve this paper.

In particular, the authors should decide what is precisely their objective: do they want to provide insights on deep C cycling or to offer a new method to compute turn-over time using 14C data? I would suspect the readers of Biogeosciences to be really interested in the first option, as there are only a limited number of studies on this topic (as claimed in l 276 of the discussion).

–> Thank you for posing this question. Indeed, our objective is to provide insights on deep soil C cycling, and not to develop a new model such as the likes of Century or RothC. We have clarified this and further simplified the modelling. We merely switch from an excel-based manual, iterative, time-consuming with limited error quantification optimization to an automated form in excel with error quantification.

Nevertheless, the data on C turn-over along the soil profile are mainly presented as supplementary, while there is a strong focus on methodological aspects in the main text.

We present the 14C data and 14C turnover data in graphs in the main texts (Figures 3-5), and the raw data can be found the SI. We have augmented our graphs.

The discussion should also be improved. Too many repetitions of the results in 4.1.1 and 4.1.2; 4.1.3 repeats some facts of 4.1.2. 4.2:

–> Thank you, we of course avoid repetition, we removed the overlapping content. The different sections do refer to the same data, so re-addressing certain patterns is unavoidable.

[Figure]

I could not find clear information in the materials and methods section about how the data supporting this section were collected.

–> Thank you, actually in sections 2.1 and 2.2 we detail that our samples are part of the long-term ecosystem monitoring program (LWF) of the Swiss Federal Institute for Forest, Snow and Landscape research, and that our ancillary data derived from publications related to this program.

The introduction/rational should refer to the needs of information on petrogenic C. 4.3: you could condense your message as you expose the same arguments for bulk C and WEOC.

–> Thank you, we have included this.

Some references to recent publications on deep C dynamics are lacking (i.e. He et al., 2016; Mathieu et al 2016; Balesdent et al 2018) while they could improve the discussion.

–> Thank you, some of these papers were already included, and we have added the rest.

I finally encourage the authors to carefully examine the relevance - and the quality - of their illustrations (see some comments below). A better focus of both the text and illustrations would guarantee a better understanding of the message the authors could deliver from the very exceptional dataset they collected.

–> Please see the comments below, indeed, visuals are key!

Some additional comments Could you indicate what is "Rsample,t" in Eq 1 and 2.

–> We have clarified this in the text

The model is based on the assumption that k1 is the turn-over of the WEOC pool. However, how do you justifythat m1 is not the size of the WEOC pool (please provide the C content of the WEOCin your MS). ?

–> Indeed, this is an assumption, we have adapted this in the text. We have included the WOEC concentration data in the Si, it is usually < 1%.

Clarify what do you mean by deep, and provide numerical value when you refer to depth in the text – currently you sometimes use it indifferently to refer to 30 cm or 80 cm, while the data strongly differ between both depths.

–> We mean > 20 cm (Mathieu et al., 2016), and have clarified this in the text.

Some Figures and Tables are offered to the readers while they are not utilised in the text: remove them (one example is Fig 3 - PS the information on the back curve is missing in the legend)

–> Thank you for noticing, we added this.

I do not understand Figure 2. How do you compute turn-over time using one individual time point?

–> We have clarified this in the text as well as the figure.

I suggest to remove Figure 5 as it is not precise – keep it for oral presentations - (what is vulnerable C?) and to provide Tables with exact numerical data in the main text.

–> Thank you, we have removed the portion about vulnerable carbon as suggested. As the heatmaps have accurate legends, we do believe it is precise enough to keep it in the paper.

Please provide the C content in for the samples measured for 14C. (Table 3 only show 3 different depths, while the data is available according to Fig 5)

–> Thank you, we have included the carbon stocks in the main text, which is most relevant when considering the turnover estimates. The carbon content data can be found in the SI as well as the Excel file with the raw data for this paper

You provide twice the particle size distribution (Tab 2 and 3).

[Figure]

–> Thank you, we have deleted the overlapping part. The difference between the tables is that Table 2 is an average Table 3 is per depth interval.

Some of your interpretations rely on soil waterlogging while this information is not clearly available (when you first mention waterlogged soil line224, the reader has not idea of which sites are concerned). In addition, I would not conclude that waterlogging is a driver of turnover by looking at the non-averaged values in Table S5.

–> Thank you, we have clarified this and adapted the interpretation.

Why are the radiocarbon signatures of WEOC different between waterlogged and non-waterlogged soils in 3.1, while calculated turnover rates are not?

–> Waterlogged soils have slower turnover, both in the bulk and in the WEOC. We have explained in the discussion that this is likely due to the impact of mineralogy as impacted by the geology, interacting with the climate.

Change your title: your gradient is not only a climatic one but a geologic one as well, with strong implication on C cycling.

–> We have highlighted the geological aspect in the introduction and discussion.

Figure 6: the colour code is not the same than in other figures.

–> Indeed, this figure shows the depth profiles dug from pits, and not the plot-averaged samples, that's why we opted for a different colour code.

I do not understand Table S1: how do you compute single resolved 14C data?

–> Thank you, this was not clear, we have clarified this in the text.

Fig S2: what stands at -20cm depth?

–> It is the 20 cm thick humus layer – we have adapted this and clarified it in the text.

Table S5: figures are not aligned in the table what makes the reading a bit tricky. The caption is not in the same order than the columns. The title of the 5th column is not clear (=> proportion of labile pool would be better)

–> Thank you for highlighting this, we have adapted Table S5 accordingly.

[Figure]

**Response to Reviewer #2**

This study aims at investigating the dynamics of carbon as a function of soil depth in five sites of the Swiss Alps. To reach this goal the authors realised 14C measurements on samples collected in late 90's and in 2014. Soils were sampled at different depths and a water extractable fraction was extracted. The authors derived C turnover rates from 14C data using a two-pool model. They identify a substantial fraction of fast-cycling C at depth and further investigate potential edaphic and climatic drivers of turnover. The data gathered in this study are of great interest, but at this stage, the manuscript suffers from too severe limitations to be published.

Thank you very much for your positive feedback regarding the quality of the dataset and insights which can be gained from it. You indicated that the main limitation was the two-pool modelling, so we addressed this, details below. We have also addressed the other issues that have been raised. Thanks again for your helpful review, it helped further improve this paper.

In particular, the authors should decide what is precisely their objective: do they want to provide insights on deep C cycling or to offer a new method to compute turn-over time using 14C data? I would suspect the readers of Biogeosciences to be really interested in the first option, as there are only a limited number of studies on this topic (as claimed in l 276 of the discussion).

Thank you for posing this question. Indeed, our objective is to provide insights on deep soil C cycling, and not to develop a new model such as the likes of Century or RothC. We have clarified this and further simplified the modelling. We merely switch from an excel-based manual, iterative, time-consuming with limited error quantification optimization to an automated form in excel with error quantification.

Nevertheless, the data on C turn-over along the soil profile are mainly presented as supplementary, while there is a strong focus on methodological aspects in the main text.

We present the 14C data and 14C turnover data in graphs in the main texts (Figures 3-5), and the raw data can be found the SI. We have augmented our graphs.

The discussion should also be improved. Too many repetitions of the results in 4.1.1 and 4.1.2; 4.1.3 repeats some facts of 4.1.2. 4.2:

Thank you, we of course avoid repetition, we removed the overlapping content. The different sections do refer to the same data, so re-addressing certain patterns is unavoidable.

I could not find clear information in the materials and methods section about how the data supporting this section were collected.

Thank you, actually in sections 2.1 and 2.2 we detail that our samples are part of the long-term ecosystem monitoring program (LWF) of the Swiss Federal Institute for Forest, Snow and Landscape research, and that our ancillary data derived from publications related to this program.

The introduction/rational should refer to the needs of information on petrogenic C. 4.3: you could condense your message as you expose the same arguments for bulk C and WEOC.

Thank you, we have included this.

Some references to recent publications on deep C dynamics are lacking (i.e. He et al., 2016; Mathieu et al 2016; Balesdent et al 2018) while they could improve the discussion.

Thank you, some of these papers were already included, and we have added the rest.

I finally encourage the authors to carefully examine the relevance - and the quality - of their illustrations (see some comments below). A better focus of both the text and illustrations would guarantee a better understanding of the message the authors could deliver from the very exceptional dataset they collected.

Please see the comments below, indeed, visuals are key!

Some additional comments
Could you indicate what is "Rsample,t" in Eq 1 and 2.

We have clarified this in the text

The model is based on the assumption that k1 is the turn-over of the WEOC pool. However, how do you justify that m1 is not the size of the WEOC pool (please provide the C content of the WEOC in your MS). ?

Indeed, this is an assumption, we have adapted this in the text. We have included the WOEC concentration data in the Si, it is usually < 1%.

**Fig. 1.**

**Dynamics of deep soil carbon – insights from [14]C time-series across a**
**climatic gradient**

Tessa Sophia van der Voort[1], Utsav Mannu[1,†], Frank Hagedorn[2], Cameron McIntyre[1,3], Lorenz Walthert[2],
Patrick Schleppi[2], Negar Haghipour[1], Timothy Ian Eglinton[1]
[1]Institute of Geology, ETH Zürich, Sonneggstrasse 5, 8092 Zürich, Switzerland
[2]Forest soils and Biogeochemistry, Swiss Federal Research Institute WSL, Zürcherstrasse 111, 8903
Birmensdorf, Switzerland
[3]Department of Physics, Laboratory of Ion Beam Physics, ETH Zurich, Schaffmattstrasse 20, 9083 Zurich
[†]New address: Department of Earth and Climate Science, IISER Pune, Pune, India
*correspondence to:* Tessa Sophia van der Voort (tessa.vandervoort@erdw.ethz.ch)

**Abstract.** Quantitative constraints on soil organic matter (SOM) dynamics are essential for comprehensive
understanding of the terrestrial carbon cycle. Deep soil carbon is of particular interest, as it represents large
stocks and its turnover rates remain highly uncertain. In this study, SOM dynamics in both the top and deep soil
across a climatic (average temperature ~1-9 °C) gradient are determined using time-series (~20 years) [14]C data
from bulk soil and water-extractable organic carbon (WEOC). Analytical measurements reveal enrichment of
bomb-derived radiocarbon in the deep soil layers on the bulk level during the last two decades. The WEOC pool
is strongly enriched in bomb-derived carbon, indicating that it is a dynamic pool. Turnover time estimates of
both the bulk and WEOC pool show that the latter cycles up to a magnitude faster than the former. The presence
of bomb-derived carbon in the deep soil, as well as the rapidly turning WEOC pool across the climatic gradient
implies that there likely is a dynamic component of carbon in the deep soil. Precipitation and bedrock type
appear to exert a stronger influence on soil C turnover and stocks as compared to temperature.

**1  Introduction**
Within the broad societal challenges accompanying climate and land use change, a better understanding of the
drivers of turnover of carbon in the largest terrestrial reservoir of organic carbon, as constituted by soil organic
matter (SOM), is essential (Batjes, 1996; Davidson and Janssens, 2006; Doetterl et al., 2015; Prietzel et al.,
2016). Terrestrial carbon turnover remains one of the largest uncertainties in climate model predictions
(Carvalhais et al., 2014; He et al., 2016). At present, there is no consensus on the net effect that climate and land
use change will have on SOM stocks (Crowther et al., 2016; Gosheva et al., 2017; Melillo et al., 2002; Schimel
et al., 2001; Trumbore and Czimczik, 2008). Deep soil carbon is of particular interest because of its large stocks
(Jobbagy and Jackson, 2000; Balesdent et al., 2018; Rumpel and Kogel-Knabner, 2011) and perceived stability.
The stability is indicated by low [14]C content (Rethemeyer et al., 2005; Schrumpf et al., 2013; van der Voort et
al., 2016) and low microbial activity (Fierer et al., 2003). Despite its importance, deep soil carbon has been
sparsely studied and remains poorly understood (Angst et al., 2016; Mathieu et al., 2016; Rumpel and Kogel-
Knabner, 2011). The inherent complexity of SOM and the multitude of drivers controlling its stability further
impedes the understanding of this globally significant carbon pool (Schmidt et al., 2011). In this framework,
there is a particular interest in the portion of soil carbon that could be most vulnerable to change, especially in
colder climates (Crowther et al., 2016). Water-exactable organic carbon (WEOC) is seen as a dynamic and
potentially vulnerable carbon pool in the soil (Hagedorn et al., 2004; Lechleitner et al., 2016). Radiocarbon
([14]C) can be a powerful tool to determine the dynamics of carbon turnover over decadal to millennial timescales

**Fig. 2.**

---

## Author Response (AR1)

Dear Sébastian,

I would like to thank you very much for supervising the revisions on my paper and for soliciting both excellent reviews from Jerome Balesdent and Reviewer #2. Both reviewers demonstrated to be experts in the field and I very much appreciate their comments. We have addressed them thoroughly and I believe this majorly improved the paper.

Overall, both reviewers were very positive about the dataset and the interpretations which they both felt merited publication. However, a major point of critique was the modelling. We acknowledge that this part of the paper was insufficiently clear, and heavily relied on the commented code which was in the Supplemental. We rectified this section following the suggestions by Mr. Balesdent and Reviewer #2, and subsequently did an appropriate overhaul of this section. We would like to clarify that we merely switched from the manual, iterative. time-consuming optimization in excel (e.g. Herold et al., 2015) to a faster, automated form in Matlab. We did not develop a new model such as the likes of CENTURY or DAYCENT. We also benchmarked all our automated results to the manual excel, and found that they agree. As reviewer #2 stressed, the focus of this paper should be on the exceptional dataset and our revisions reflect this. We have also added all additional requested information such as quantified residual errors after the optimization.

In accordance with the submission guidelines, the code and datasets will be deposited in FAIR-aligned data repositories. For the review process, we have also included the Matlab codes for the reviewers in the supplement.

Please find detailed replies to comments in our point-by-point replies to the reviewers.

Thank you again very much,

Kind regards on behalf of all my co-authors,

Tessa

**Response to Jerome Balesdent**

Understanding the dynamics of carbon in deep soil layers is an important issue, and this study uses an excellent sequence and provides a rare dataset: soil 14C measure- ment at two dates using archived samples brings a precious information of C dynamics. One of the interesting results is the demonstration of the occurrence of rock-derived carbon. Another concerns the age of water extractable carbon. The analytical meth- ods are high standard and highly relevant. I therefore consider it is worth publishing the data in Biogeochemistry. Unfortunately, there are major concerns that need revi- sion. The most important is that the mathematical and numerical interpretations look inappropriate, and this leads the authors to give conclusions that are in contrast with what the data show, whereas some unprecedented results could be derived. I finally suggest two alternative solutions: either the authors drop the modelling part and make a semi-quantitative interpretation of the data, either they use another model. I also noticed miscellaneous improvements to be done. The discussion should be updated according to these major points. The title and summary are nevertheless appropriate.

Dear Prof. Balesdent,

Thank you very much for your positive feedback and thorough review. We very much appreciate that you value the importance of the data for the wider Biogeosciences community. Your comments about the turnover time modelling are also very insightful and the issues have subsequently been addressed. There was indeed a semantic issue which caused problems, so we incorporated all of your feedback. We realized that most of the modelling was explained in the code in the SI, and that therefore the text in the main text was absolutely inadequate in order to explain our calculations. Consequently, paper and especially the discussion was updated according to these major points. As you indicated, the title and summary remained appropriate.

We want to thank you again for your helpful review, which has further improved this paper. Please find detailed replies below.

1. The chosen model is unlikely to simulate observed data.

Most of samples below 10 cm show an increase in Δ14C between 1990's and 2010's, by several 10‰ (Figure 3), and even some above 10 cm do. As seen in the FIGURE below, which was built for this review, the 14C content of well mixed compartments directly fed from atmospheric C has DECREASED with time since the 1990's (or in- creased by less than 4‰ for slow pools). The sum of two parallel pools cannot have a Δ14C increased between 1995 and 2014.

FIGURE: Simulated Δ14C of a well-mixed compartment under steady state as a func- tion of compartment turnover rate, for two dates of sampling.

Thank you for these comments. Indeed, rapidly turning-over compartments have decreased in [14]C in the last two decades, whilst the slower compartments have increased in [14]C signature (e.g. Figure 3a) (as you also indicate in your comments below). As you indicated in your supplemental, there was indeed a semantic issue with the turnover time definition when estimating the size of the two respective pools, which has now been adapted. Our apologies for the confusion.

I finally understood (from 14C data in Figure 3 and turnover time data in Table S5) that the the "mosty reliable' kWSOC value is more or less the arithmetic mean of two kWSOC values, one calculated in the 1990's and the other in the 2010's. The authors must invoke other processes to explain an increasing Δ14C. These processes may act together and interact:

- Transit of carbon in another horizons or pool before entering the observed layer. This might be associated with either bioturbation or DOC production from an above layer, movement, and insolubilization. The data tend to indicate that carbon movement is a significant cause of the increase in Δ14C across the sequence.

- non-steady state, e.g. increased bioturbation due to warming, change in NPP and/or decay rates.

To me, the fact that the Δ14C of WSOC of all samples (except Othmarsingen 0-5 cm and Lausanne 0-5 cm) inceased is a proof that WSOC is a by-product of SOM aged several 10th of years (usual age of OH horizons), and not directly fed by vegetation decomposition. This would be a bright finding and merit appropriate modelling.

Thank you for these comments as well. We did not sufficiently explain how we estimate the turnover of the WEOC or 'labile' pool using the $^{14}$C time-series. We have addressed this now in the method section, by detailing the different steps and the error calculation (Equations 1-4). In short, we do not take the arithmetic mean, but rather use the standard equations (e.g. Herold et al., 2014; Torn et al., 2009) to find the likeliest turnover time considering both data points of the time series. Instead of the usual excel-based method, we do this in MatLab because it is automated and more repeatable. The solution which has the lowest calculated residual square root mean error (RSME) is automatically chosen, as opposed to a manual iteration.

Thank you for highlighting the importance of potentially DOC-driven transport of young(er) carbon through the deep soil, we have included this in our discussion. We have also now included your suggestion in Section 4.1.3 to highlight that WEOC is likely not fed by vegetation decomposition but rather is derived from several decades-old SOM.

2. Consistency in model implementation (to be confirmed).

I tried to calculate by myself turnover time values, based on 14C data in Figure 3 and turnover time data in Table S5, and didn't find the author's results. This may arise from the fact that the basic differential equations of the model (equation 5 = SI.7) looks false, or at least do not correspond to authors' hypotheses. Equation SI.7 states:

$F(t) = k \cdot F_{atm}(t) + m1 \cdot F(t - 1).(1 - \lambda - k1) + m2 \cdot F(t-1).(1 - \lambda - k2)$
This equation indicates that the flux of 14C leaving the system (out of desintegration)

is:

$(m1.k1 + m2.k2).F(t -1)$, i.e., $k.F(t)$

Since the corresponding flux of carbon is $k = m1.k1 + m2.k2$, this equation says that the 14C activity of carbon leaving the system is $F(t - 1)$. So the equation would IM- PLICITELY considers that the activity of the flux out is the same as that of the compart- ment itself. This is typically the assumption of a so-called 'well mixed' compartment, and is not the case of a system with two compartments. It would only accept the solu- tion $k1 = k2$. Making this implicit assumption is a current mistake or at least a source of disagreement in isotope geochemisty. As a consequence, I guess that the authors have calculated a mean turnover time corresponding to a single compartment for bulk carbon, and an independent specific turnover time of WSOC. The error might be linked with my point 3 below. See a proposal for the correct equation as an appendix of this review. The authors are invited to check how eq SI.7 was implemented and how the couple $(k2, m1)$ was inferred from bulk F14C.

Thank you, there are two main things raised in this comment:

A. Modelling Structure

Indeed, we have calculated a mean turnover time corresponding to a single compartment for bulk carbon, and one independent specific turnover time of WSOC. We have clarified this in the text.

B. Model consistency

Thank you for your suggestions and example for Figure SI.7, we have implemented all of your suggestions (Eq. 6). More details can be found below.

We would like to clarify that we merely transformed the usual excel file people use to find turnover time to MatLab-driven optimization, because it saves time, is repeatable, unbiased and error can be quantified. We have now also quantified all our errors (See SI). Furthermore, the code can easily be used as well for longer time-series (i.e. > 2 timepoints). We benchmarked our results to the Excel-based method, and the results agree.

3. Mathematical (and semantic) misuse of "turnover time'.

Let us call the turnover time of carbon in the compartment $T = 1/k$ Mathematically, the carbon input to the system is $m1/T1 + m2/T2$. The size of the compartment is $m1 + m2$. So, the turnover time, which is the ratio of pool size to the input, is:

$T = (m1 + m2)/(m1/T1 + m2/T2)$

In Table SI.5, which presents the main result, i.e. the values of turnover time, the authors calculated the bulk turnover time as:

$T = (m1.T1 + m2.T2)/(m1 + m2)$, which is wrong.

What authors call "turnover time" is in fact the MEAN AGE of carbon, which is different of the mean turnover time in non-well mixed compartments. The error in not only semantic because it possibly have interfered in model and 14C equation (point 2). Sierra et al. (2016), whom you cite lines 161-162, recommends the use of "age", not "turnover time" for this variable. See also Manzoni et al.(2009).

Indeed, there was a (semantic) inconsistency regarding turnover time between the Main text and the SI, which we have now addressed and corrected. We also implemented your equation. We have the [14]C-determined 'turnover time' for the bulk soil, whilst stating that we assume a steady state. We have also clarified our definition in the text, following Manzoni et al. (2009) as well.

4. Data availability.

The authors must provide in SI a table including the primary data, i.e., Δ14C, C stock by horizon, WEOC stocks. Reference that were used to estimate atmospheric Δ14C (post bomb and pre-bomb) should be indicated (e.g. Reimer , Hua etc.)

We have included an excel file with all the raw data regarding $\Delta^{14}$C and stocks the WEOC material. The WEOC concentration is low (< 1 %) and can be found in SI Table 4. We had indicated the provenance of our pre- and post-bomb data already in the method section, but we have now further clarified it.

5. Hypothesis on WSOC as the labile pool.

Line 180-182 and 190-191: A major (if not the major) assumption of the model is that the dynamic pools has the same decay rate as that of WEOC. The 'dynamic' pools contains as much as 88% of soil C (on the average 34%), whereas WEOC only a few %. Assigning the constant k of WEOC to the dynamic pool is therefore a surprising and very heavy hypothesis. (see also point 1.)

Alternatively, the study may have targetted the study of WSOC dynamics for itself, e.g., considered that both WSOC and bulk C are heterogenous pools, each with a labile and a more stable component, but in varied proportion. Many other models use particulate organic matter (i.e. either sand-size primary organic particles or light OM, which has been described as having a good fit with labile carbon

Yes indeed, it was our assumption that the measured WEOC could be representative of the dynamic pool. There are studies that hypothesize WEOC could be indicative of a larger dynamic pool (Baisden and Parfitt, 2007; Koarashi et al., 2012). But indeed, this is a heavy assumption. We have therefore decreased the importance of the two-pool model in the paper, and highlighted this assumption. Indeed, both the WEOC and bulk themselves can heterogeneous pools, hence we also looked at biomarkers in another study (e.g. Van der Voort et al., 2017, Diverse Soil Carbon Dynamics Expressed at the Molecular Level, GRL). Looking at other fraction would be a worthwhile topic for future work.

6. Conclusions on correlation with MAP.

Projecting conclusions on the effect of MAP on the basis of a "wet" sequence, i.e., where the water deficit is probably low if not nil, may look brash. The driest site is 800 mm, but with a MAT 1.3C and probably a small PET. Furthermore (Lines 360-361), authors state that 'The only climate-related driver which appears to be significant is precipitation' whereas the r2 coefficient between MAP and turnover 0-20 cm is 0.04! I would recommend here to cite Carvalhais et al. (2013) and Mathieu et al. (2015), who highlighted the role of precipitation in SOM stabilization or ecosystem carbon turnover.
I sfinally uggest to moderate the conclusions, but maybe discuss the role of precipitation on DOC movement (see point 1).

Thank you for these insights. As suggested, we have highlighted the role of precipitation as SOM stabilizer and interaction with DOC movement, and tempered our statements about precipitation. Indeed, Switzerland is a wet country! Your own 2018 paper also could show the important role of evapotranspiration but we unfortunately do not have this data. Also, we adapted the phrasing of line 360-61, the role of precipitation is pronounced for the deep soil.

7. Presentation of model and equations.

The presentation of both the model and the optimization process is obscure throughout the text and should be more precise, in either text or SI. In the cases with four radio- carbon dates (2 sampling dates x two fractions), the optimization of three dynamic pa- rameters is not a formal solution, but a best fit.

Indeed, we have now mentioned this specifically in the text.

The type of adjustment (least squares ?) and a criterion of the fit (e.g., RMSE) should be indicated.

This has been included in the main text instead of the SI, we use RSME.

Harmonize the name of variables throughout the text and SI. For consistency with SI, please use m insteaf of F in eqn (3), (4) and (4); and possibly F instead of R. Also use the same character k in SI and main text. Harmonize M (Figure S2) and m, etc.

Thank you, this has now been adjusted.

How were single points managed ? (Line 194-195. " Due to limited availability of archived samples, there are only single time points available for some samples as indicated in Fig. 4.")

This has been clarified in the main text, we solve the standard radiocarbon decay equations (e.g. Torn et al., 2009, $R_{sample,t} = k \times R_{atm,t} + (1 - k - \lambda) \times R_{sample(t-1)}$ ). This is done more traditionally in Excel, we did the same using a Matlab optimization.

8. Miscellaneous.

lines 51-52 note the pioneer studies by Jenkinson et al (1992) on long-term experi- ments. The models by Braakkeke et al. (2014 ) also simulates 14C profiles in rather similar podzols, using WSOC as well, and may receive more attention in the discussion section. Also note (e.g. Line 34) the conclusions of Mathieu et al. (2015) concerning soil versus climate drivers of 14C, and (lines 39-40) the recent paper by Balesdent et al. (2018), which improved the understanding of the significance of deep soil C to the global C cycle.

Thank you, I have incorporated these literature suggestions. I had already cited Braakhekke et al.

Move lines 126-128 (WEOC) to the end of 2.1. (WEOC extraction). Note that extraction with Na 0.86 M is not exactly Water extraction, since it moves some exchangeable calcium, disperses clays and therefore moves sorbed organic compounds that would not have been mobilized by water.

Indeed, we followed Hagedorn et al., 2014 when preparing the extraction, and have this stated this clearly in the method section.

Line 252 ' Deeper soil bulk stock and turnover positively...' and table S5: avoid "turnover " alone standing for "turnover time" in such sentences, because the common sense of turnover is turnover rate, i.e., the inverse of turnover time. This may lead to a reverse understanding of correlations.

Indeed! We adapted this now.

Line 262. Balesdent et al. (2018) reported that 21% of world subsoil C (30-100 cm) is less than 50 years old.

We have included this.

The amount of WEOC (while not used in the modelling experiment) would be welcome.

We have included this in the SI Table 4. Concentrations are low (< 1%)

Surprisingly, the section of Material and methods indicates that NPP and its compo- nents were measured, which is a rare information in SOM studies. As a result, authors have an indicator of the true turnover time of soil C, i.e. the ratio of Soil C stock to C input is known, that they do not use.

Indeed, there is NPP data, but we were recommended by the field experts that although it was representative for the tree vegetation, we had better not use it for estimating soil flux, as there would be too many assumptions to be considered. We did include the data, so others are free to use it.

Figure 4 contains the main primary result of the study. Polices Should be enlarged. The square signs for Aptal WEOC 1997 are misleading. Table S5 is the main final result and should take place in the main document.

We have adapted this figure slightly. Following your critique about the assumption of using WEOC as a dynamic pool we reduced the importance of the fraction modelling in the paper, so we opted to keep it in the SI.

Note that the bi-exponential age distribution is factually the age distribution of C in current "four pools" models such as RothC (or Century). All coupling of these models with radiocarbon more or less managed bi-exponential age distribution and 14C; e.g., Jenkinson et al. (1992).

Yes, we are familiar with Century (RothC), but feel applying them would be beyond the scope of this paper.

9. Appendix

The differential equation should consider F1 and F2 the 14C fraction in pools 1 and 2, respectively, as illustrated in your Fig S1.

Input flux to pool1 is k1.m1; input flux to pool2 is k2.m2

F1(t) =k1.Fatm(t) + (1 - k1 - λ).F1(t - 1)  F2(t) =k2.Fatm(t) + (1 - k2 - λ).F2(t - 1)  which give: F(t) = m1F1(t) + m2.F2(t) = k.Fatm(t) + m1.(1 - k1 - λ).F1(t - 1) + m1.(1 – k2 - λ).F2(t - 1) And needs numerical resolution of F1 and F2.

Thank you, we implemented this.

10. Cited references Balesdent J., Basile-Doelsch I, Chadoeuf J., Cornu S., Derrien D. Fekiacova Z., Hatté C. Atmosphere-soil carbon transfer as a function of soil depth. Nature, 559, 599–602. (2018) doi.org/10.1038/s41586-018-0328-3

Jenkinson D.S., D.D. Harkness, E.D. Vance, D.E. Adams and A.F. Harrison. Calculating net primary production and annual input of organic matter to soil from the amount and radiocarbon content of soil organic matter. Soil Biol. Biochem. 24(4):295-308 (1992)

Manzoni, S., Katul, G. G. & Porporato, A. Analysis of soil carbon transit times and age distributions using network theories. J. Geophys. Res. 114, G04025 (2009)

Mathieu J., Hatté C., Parent E., Balesdent J. Deep soil carbon dynamics are driven more by soil type than by climate: a worldwide meta-analysis of radiocarbon profiles. Global Change Biology 21, 4278-4292. (2015) doi:10.1111/gcb.13012.

Thank you, we implemented these papers

11. Figure.

Simulated Δ14C of a well-mixed compartment under steady state as a function of compartment turnover rate, for two dates of sampling. Compartment has a single C7

exponential distribution of ages; system start 8050 BP; atmospheric Δ14C after Reimer et al. (2009) and Hua et al. (2013); Northern hemisphere zone N2; May-August.

**Response to Reviewer #2**

This study aims at investigating the dynamics of carbon as a function of soil depth in five sites of the Swiss Alps. To reach this goal the authors realised 14C measurements on samples collected in late 90's and in 2014. Soils were sampled at different depths and a water extractable fraction was extracted. The authors derived C turnover rates from 14C data using a two-pool model. They identify a substantial fraction of fast-cycling C at depth and further investigate potential edaphic and climatic drivers of turnover. The data gathered in this study are of great interest, but at this stage, the manuscript suffers from too severe limitations to be published.

*Thank you very much for your positive feedback regarding the quality of the dataset and insights which can be gained from it. You indicated that the main limitation was the two-pool modelling, so we addressed this, details below. We have also addressed the other issues that have been raised. Thanks again for your helpful review, it helped further improve this paper.*

In particular, the authors should decide what is precisely their objective: do they want to provide insights on deep C cycling or to offer a new method to compute turn-over time using 14C data? I would suspect the readers of Biogeosciences to be really interested in the first option, as there are only a limited number of studies on this topic (as claimed in l 276 of the discussion).

*Thank you for posing this question. Indeed, our objective is to provide insights on deep soil C cycling, and not to develop a new model such as the likes of Century or RothC. We have clarified this and further simplified the modelling. We merely switch from an excel-based manual, iterative, time-consuming optimization with limited error quantification to an automated form in excel with error quantification.*

Nevertheless, the data on C turn-over along the soil profile are mainly presented as supplementary, while there is a strong focus on methodological aspects in the main text.

*We present the $^{14}$C data and $^{14}$C turnover data in graphs in the main texts (Figures 3-5), and the raw data can be found the SI. We have augmented our graphs.*

The discussion should also be improved. Too many repetitions of the results in 4.1.1 and 4.1.2; 4.1.3 repeats some facts of 4.1.2. 4.2:

*Thank you, we of course avoid repetition, we removed the overlapping content. The different sections do refer to the same data, so re-addressing certain patterns is unavoidable.*

I could not find clear information in the materials and methods section about how the data supporting this section were collected.

*Thank you, actually in sections 2.1 and 2.2 we detail that our samples are part of the long-term ecosystem monitoring program (LWF) of the Swiss Federal Institute for Forest, Snow and Landscape research, and that our ancillary data derived from publications related to this program.*

The introduction/rational should refer to the needs of information on petrogenic C. 4.3: you could condense your message as you expose the same arguments for bulk C and WEOC.

*Thank you, we have included this.*

Some references to recent publications on deep C dynamics are lacking (i.e. He et al., 2016; Mathieu et al 2016; Balesdent et al 2018) while they could improve the discussion.

*Thank you, some of these papers were already included, and we have added the rest.*

I finally encourage the authors to carefully examine the relevance - and the quality - of their illustrations (see some comments below). A better focus of both the text and illustrations would guarantee a better understanding of the message the authors could deliver from the very exceptional dataset they collected.

*Please see the comments below, indeed, visuals are key!*

Some additional comments
Could you indicate what is "Rsample,t" in Eq 1 and 2.

*We have clarified this in the text*

The model is based on the assumption that k1 is the turn-over of the WEOC pool. However, how do you justifythat m1 is not the size of the WEOC pool (please provide the C content of the WEOCin your MS). ?

*Indeed, this is an assumption, we have adapted this in the text. We have included the WOEC concentration data in the Si, it is usually < 1%.*

Clarify what do you mean by deep, and provide numerical value when you refer to depth in the text – currently you sometimes use it indifferently to refer to 30 cm or 80 cm, while the data strongly differ between both depths.

We mean > 20 cm (Mathieu et al., 2016), and have clarified this in the text.

Some Figures and Tables are offered to the readers while they are not utilised in the text: remove them (one example is Fig 3 - PS the information on the back curve is missing in the legend)

Thank you for noticing, we added this.

I do not understand Figure 2. How do you compute turn-over time using one individual time point?

We have clarified this in the text as well as the figure.

I suggest to remove Figure 5 as it is not precise – keep it for oral presentations - (what is vulnerable C?) and to provide Tables with exact numerical data in the main text.

Thank you, we have removed the portion about vulnerable carbon as suggested. As the heatmaps have accurate legends, we do believe it is precise enough to keep it in the paper.

Please provide the C content in for the samples measured for 14C. (Table 3 only show 3 different depths, while the data is available according to Fig 5)

Thank you, we have included the carbon stocks in the main text, which is most relevant when considering the turnover estimates. The carbon content data can be found in the SI as well as the Excel file with the raw data for this paper

You provide twice the particle size distribution (Tab 2 and 3).

Thank you, we have deleted the overlapping part. The difference between the tables is that Table 2 is an average Table 3 is per depth interval.

Some of your interpretations rely on soil waterlogging while this information is not clearly available (when you first mention waterlogged soil line224, the reader has not idea of which sites are concerned). In addition, I would not conclude that waterlogging is a driver of turnover by looking at the non-averaged values in Table S5.

Thank you, we have clarified this and adapted the interpretation.

Why are the radiocarbon signatures of WEOC different between waterlogged and nonwaterlogged soils in 3.1, while calculated turnover rates are not?

Waterlogged soils have slower turnover, both in the bulk and in the WEOC. We have explained in the discussion that this is likely due to the impact of mineralogy as impacted by the geology, interacting with the climate.

Change your title: your gradient is not only a climatic one but a geologic one as well, with strong implication on C cycling.

We have highlighted the geological aspect in the introduction and discussion.

Figure 6: the colour code is not the same than in other figures.

Indeed, this figure shows the depth profiles dug from pits, and not the plot-averaged samples, that's why we opted for a different colour code.

I do not understand Table S1: how do you compute single resolved 14C data?

Thank you, this was not clear, we have clarified this in the text.

Fig S2: what stands at -20cm depth?

It is the 20 cm thick humus layer – we have adapted this and clarified it in the text.

Table S5: figures are not aligned in the table what makes the reading a bit tricky. The caption is not in the same order than the columns. The title of the 5th column is not clear (=> proportion of labile pool would be better)

Thank you for highlighting this, we have adapted Table S5 accordingly.

[revised manuscript text omitted]

Font:Not Bold

| Page 5: [1] Formatted | Tessa Sophia | 2/3/19 2:51:00 PM |
|---|---|---|

Font:Not Bold

| Page 5: [1] Formatted | Tessa Sophia | 2/3/19 2:51:00 PM |
|---|---|---|

Font:Not Bold

| Page 5: [1] Formatted | Tessa Sophia | 2/3/19 2:51:00 PM |
|---|---|---|

Font:Not Bold

| Page 5: [1] Formatted | Tessa Sophia | 2/3/19 2:51:00 PM |
|---|---|---|

Font:Not Bold

| Page 5: [1] Formatted | Tessa Sophia | 2/3/19 2:51:00 PM |
|---|---|---|

Font:Not Bold

| Page 5: [2] Deleted | Tessa Sophia | 1/30/19 6:48:00 PM |
|---|---|---|

For the turnover estimation, we assumed the system to be in steady state over the modeled period ($\sim 1\times10^4$ years, indicating soil formation since the last glacial retreat (Ivy-Ochs et al., 2009)), hence accounting both for radioactive decay and incorporation of the bomb-testing derived material produced in the 1950's and 1960's (Eq. 1.) (Herold et al., 2014; Torn et al., 2009).

**2.4.1 Time-series based determination of likeliest turnover time**

In order to optimally constrain carbon turnover estimates for the [14]C time-series data, a numerical model was constructed in MATLAB version 2015a (The MathWorks, Inc., Natick, Massachusetts, United States). For the turnover estimation, we assumed the system to be in steady state over the modeled period ($\sim 1\times10^4$ years, indicating soil formation since the last glacial retreat (Ivy-Ochs et al., 2009)), hence accounting both for radioactive decay and incorporation of the bomb-testing derived material produced in the 1950's and 1960's (Eq. 1.) (Herold et al., 2014; Torn et al., 2009).

| Page 5: [3] Formatted | Tessa Sophia | 2/8/19 1:42:00 PM |
|---|---|---|

Font:10 pt, Italic

| Page 5: [3] Formatted | Tessa Sophia | 2/8/19 1:42:00 PM |
|---|---|---|

Font:10 pt, Italic

| Page 5: [3] Formatted | Tessa Sophia | 2/8/19 1:42:00 PM |
|---|---|---|

Font:10 pt, Italic

| Page 5: [3] Formatted | Tessa Sophia | 2/8/19 1:42:00 PM |
|---|---|---|

Font:10 pt, Italic

| Page 5: [4] Formatted | Tessa Sophia | 2/3/19 2:51:00 PM |
|---|---|---|

Font:10 pt

| Page 5: [4] Formatted | Tessa Sophia | 2/3/19 2:51:00 PM |
|---|---|---|

Font:10 pt

| Page 5: [5] Formatted | Tessa Sophia | 2/8/19 1:43:00 PM |
|---|---|---|

Font:10 pt

| Page 5: [6] Formatted | Tessa Sophia | 2/3/19 2:51:00 PM |
|---|---|---|

Font:10 pt

| Page 5: [7] Formatted | Tessa Sophia | 2/8/19 3:49:00 PM |
|---|---|---|

Font:10 pt

| Page 5: [8] Formatted | Tessa Sophia | 2/3/19 2:51:00 PM |
|---|---|---|

Superscript

| Page 5: [8] Formatted | Tessa Sophia | 2/3/19 2:51:00 PM |
|---|---|---|

Superscript

| Page 5: [8] Formatted | Tessa Sophia | 2/3/19 2:51:00 PM |
|---|---|---|

Superscript

| Page 5: [8] Formatted | Tessa Sophia | 2/3/19 2:51:00 PM |
|---|---|---|

Superscript

| Page 6: [9] Moved to page 5 (Move #3) | Tessa Sophia | 1/30/19 7:50:00 PM |
|---|---|---|

Turnover times determined with the numerical optimization match the manually optimized turnover modeling published previously (Herold et al., 2014; Solly et al., 2013).

| Page 6: [10] Formatted | Tessa Sophia | 2/8/19 3:56:00 PM |
|---|---|---|

List Paragraph, Justified, Outline numbered + Level: 3 + Numbering Style: 1, 2, 3, ... + Start at: 1 + Alignment: Left + Aligned at:  0 cm + Indent at:  1.27 cm

| Page 6: [11] Deleted | Tessa Sophia | 2/3/19 1:05:00 PM |
|---|---|---|

 (Fig. 3[TSvdV1]). WEOC constitutes only a small portion of the total carbon (<1%), but could be representative for a larger component of rapidly turning over carbon, even in the deep soil (Baisden and Parfitt, 2007; Koarashi et al., 2012). Using the data from the bulk soil and WEOC time-series, the turnover of the slow pool and the relative size of the dynamic pool can be determined.

| Page 7: [12] Deleted | Tessa Sophia | 2/3/19 4:31:00 PM |
|---|---|---|

Where $F_1$ is the relative size of the dynamic pool, and $F_2$ is the relative size of the (more) stable pool. The $k_1$is [TSvdV2]the inverse of the turnover time of the WEOC as determined using the numerical optimisation of Eq. (1) and (2), and $k_1$ is determined by numerical optimisation. The $k_2$ is the inverse of the turnover of the slow pool.

| Page 21: [13] Formatted | Tessa Sophia | 2/3/19 2:51:00 PM |
|---|---|---|

Font:10 pt

| Page 21: [14] Formatted | Tessa Sophia | 2/3/19 4:30:00 PM |
|---|---|---|

Justified

| Page 21: [15] Formatted | Tessa Sophia | 2/3/19 4:30:00 PM |
|---|---|---|

Justified

| Page 21: [16] Formatted | Tessa Sophia | 2/3/19 4:30:00 PM |
|---|---|---|

Justified

| Page 21: [17] Formatted | Tessa Sophia | 2/3/19 4:30:00 PM |
|---|---|---|

Justified

| Page 21: [18] Formatted | Tessa Sophia | 2/3/19 4:30:00 PM |
|---|---|---|

Justified

| Page 21: [19] Formatted | Tessa Sophia | 2/3/19 4:30:00 PM |
|---|---|---|

Justified

| Page 21: [20] Formatted | Tessa Sophia | 2/3/19 4:30:00 PM |
|---|---|---|

Justified

| Page 21: [21] Formatted | Tessa Sophia | 2/3/19 4:30:00 PM |
|---|---|---|

Justified

| Page 21: [22] Deleted | Tessa Sophia | 2/6/19 1:58:00 PM |
|---|---|---|
| Page 21: [22] Deleted | Tessa Sophia | 2/6/19 1:58:00 PM |

-

-

| Page 21: [22] Deleted | Tessa Sophia | 2/6/19 1:58:00 PM |
|---|---|---|
| Page 21: [23] Deleted | Tessa Sophia | 2/6/19 2:16:00 PM |

-

-

| Page 21: [23] Deleted | Tessa Sophia | 2/6/19 2:16:00 PM |
|---|---|---|
| Page 21: [24] Deleted | Tessa Sophia | 2/6/19 2:22:00 PM |

-

-

| Page 21: [24] Deleted | Tessa Sophia | 2/6/19 2:22:00 PM |
|---|---|---|
| Page 21: [24] Deleted | Tessa Sophia | 2/6/19 2:22:00 PM |

-

-

| Page 21: [25] Formatted | Tessa Sophia | 2/3/19 4:30:00 PM |
|---|---|---|

Justified

| Page 21: [26] Formatted | Tessa Sophia | 2/3/19 4:30:00 PM |
|---|---|---|

Justified

| Page 21: [27] Deleted | Tessa Sophia | 2/6/19 1:58:00 PM |
|---|---|---|
| Page 21: [27] Deleted | Tessa Sophia | 2/6/19 1:58:00 PM |

-

-

| Page 21: [28] Formatted | Tessa Sophia | 2/3/19 4:30:00 PM |
|---|---|---|

Justified

| Page 21: [29] Formatted | Tessa Sophia | 2/3/19 4:30:00 PM |
|---|---|---|

Justified

| Page 21: [30] Formatted | Tessa Sophia | 2/3/19 4:30:00 PM |
|---|---|---|

Justified

| Page 21: [31] Formatted | Tessa Sophia | 2/3/19 2:51:00 PM |
|---|---|---|

Font:10 pt

| Page 21: [32] Formatted | Tessa Sophia | 2/3/19 4:30:00 PM |
|---|---|---|

Justified

---

## Author Response (AR2)

Dear Sébastian,

**Thank you** for going through the referee's comments and for your additional suggestions. You suggested to include soil type in the figure labeling and to clarify the counterintuitive turnover pattern in the Podzol. We have **incorporated** them:

(1) **In the figures, we have now included both the soil type and location.** We have also taken the opportunity to improve the colour scheme of Figure 4.

(2) **Thank you for highlighting the Podzol dynamics.** Indeed, you are right. Oxides and hydroxides are crucial for stabilization, so higher C dynamics in the deep soil versus the shallower soil is counterintuitive. We did not sufficiently explain the pedogenetic horizons in the Podzol in order to explain the pattern. The crux is the transition from the elluvial to the illuvial horizon. We have now corrected this.
**In detail:**
The elluvial horizon in Beatenberg extends from 4-5 cm downwards up to ~35 cm. The illuviation leeches the iron, aluminium out of the soil layer. Deeper, in the illuviation horizon (~35-60 cm), there is a redeposition of iron and aluminium which stabilizes the organic matter.
The counterintuitive pattern that you correctly noted is that in the shallower 20-40 cm layer - which encompasses the elluvial horizon – the turnover is slower than the deeper (40-60) layer - which encompasses the illuvial soil horizon. The reason the counter-intuitive pattern occurs is that elluvial horizon (mostly represented in the 20-40 layer) there are lower amounts of oxydes and hydroxydes as compared to the deeper layers. In the deeper layer (40-60), the organic matter can form organo-mineral complexes again and the carbon dynamics are faster.

Thank you very much again, we look forward to the feedback of the referees,

Tessa

[revised manuscript text omitted]

**Comment [TSvdV6]:** Dear Sébastian, as suggested we have now included the soil types both in the figure heading as well as the text. We have improved the text following this suggestion elsewhere as well. We have also augmented the colours in this figure as compared to the previous version.

[Figure]

**Figure 5** Carbon (a) stocks in the mineral soil kgC/m$^2$, (b) turnover time bulk soil in years and (c) turnover time water extractable organic carbon soil in years. Locations are ordered from the warmest to coldest sites i.e. (1) Othmarsingen (Luvisol), (2) Lausanne (Cambisol), (3) Alptal (Gleysol), (4) Beatenberg (Podzol) and (5) Nationalpark (Fluvisol). Grey boxes indicate absence of material, black boxes indicate the occurence of the C-horizon (poorly consolidated bedrock-derived stony material or bedrock itself).

**Comment [TSvdV7]:** Figure also augmented with soil type as suggested

[Figure]

**Figure 6** Modeled turnover times (y) of single profiles sampled down to the bedrock between 1995 and 1998. $\Delta^{14}$C published in Van der Voort et al. (2016). Results indicate presence of petrogenic (bedrock-derived) carbon as modeled turnover time exceeds soil formation since the end of last ice age (10,000 years) in Lausanne (>100 cm, Cambisol) and Alptal (80-100 cm, Gleysol). For Beatenberg (Podzol) and Nationalpark (Fluvisol), no petrogenic carbon was found.

---

## Author Response (AR3)

**Point by point replies**

Dear Sébastian, dear reviewer,

All suggested minor revisions were incorporated. The reviewer had some valuable suggestions regarding word-use and terminology, and phrasing was adjusted were needed. We also updated some affiliations and sample storage details.

We would very much like to thank the reviewers and the editor again for improving this paper, we really appreciate their time.

Kind regards on behalf of all co-authors,

Tessa

[revised manuscript text omitted]